# Expected Sliced Transport Plans

**Xinran Liu**[*1], **Rocío Martín Díaz**[*2], **Yikun Bai**[1], **Ashkan Shahbazi**[1],
**Matthew Thorpe**[3], **Akram Aldroubi**[4], **Soheil Kolouri**[1]

[1]Department of Computer Science, Vanderbilt University, Nashville, TN, 37235
[2]Department of Mathematics, Tufts University, Medford, MA 02155
[3]Department of Statistics, University of Warwick, Coventry, CV4 7AL, UK
[4]Department of Mathematics, Vanderbilt University, Nashville, TN, 37235

## Abstract

The optimal transport (OT) problem has gained significant traction in modern machine learning for its ability to: (1) provide versatile metrics, such as Wasserstein distances and their variants, and (2) determine optimal couplings between probability measures. To reduce the computational complexity of OT solvers, methods like entropic regularization and sliced optimal transport have been proposed. The sliced OT framework improves efficiency by comparing one-dimensional projections (slices) of high-dimensional distributions. However, despite their computational efficiency, sliced-Wasserstein approaches lack a transport plan between the input measures, limiting their use in scenarios requiring explicit coupling. In this paper, we address two key questions: Can a transport plan be constructed between two probability measures using the sliced transport framework? If so, can this plan be used to define a metric between the measures? We propose a 'lifting' operation to extend one-dimensional optimal transport plans back to the original space of the measures. By computing the expectation of these lifted plans, we derive a new transport plan, termed expected sliced transport (EST) plans. We further prove that using the EST plan to weight the sum of the individual Euclidean costs $\|x-y\|^p$ for moving from $x$ to $y$ results in a valid metric between the input discrete probability measures. Finally, we demonstrate the connection between our approach and the recently proposed min-SWGG, along with illustrative numerical examples that support our theoretical findings.

## 1 Introduction

The optimal transport (OT) problem (Villani, 2009) seeks the most efficient way to transport a distribution of mass from one configuration to another, minimizing the cost associated with the transportation process. It has found diverse applications in machine learning due to its ability to provide meaningful distances, i.e., the Wasserstein distances (Peyré & Cuturi, 2019), between probability distributions, with applications ranging from supervised learning (Frogner et al., 2015) to generative modeling (Arjovsky et al., 2017). Beyond merely measuring distances between probability measures, the optimal transport plan obtained from the OT problem provides correspondences between the empirical samples of the source and target distributions, which are used in various applications, including domain adaptation (Courty et al., 2014), positive-unlabeled learning (Chapel et al., 2020), texture mixing (Rabin et al., 2011), color transfer (Rabin et al., 2014), image analysis (Basu et al., 2014), and even single-cell and spatial omics (Bunne et al., 2024), to name a few.

One of the primary challenges in applying the OT framework to large-scale problems is its computational complexity. Traditional OT solvers for discrete measures typically scale cubically with the number of samples $N$ (i.e., the support size) (Kolouri et al., 2017). Precisely, when using linear programming for solving the OT problem, the computational complexity is of order $\mathcal{O}(N^3 \log(N))$. This computational burden has spurred significant research efforts to accelerate OT computations. Various approaches have been developed to address this challenge, including the seminal work (Cuturi, 2013), which introduces entropic regularization, leveraging the Sinkhorn algorithm to compute OT efficiently with quadratic time complexity; multiscale methods (Schmitzer, 2016); and projection-

---

[*]Equal contribution.

based techniques such as sliced-Wasserstein distances (Rabin et al., 2011) and robust subspace OT (Paty & Cuturi, 2019). Each of these methods has its own advantages and limitations.

For instance, the entropic regularized OT is solved via an iterative algorithm (i.e., the Sinkhorn algorithm) with quadratic computational complexity per iteration. However, the number of iterations required for convergence typically increases as the regularization parameter decreases, which can offset the computational benefits of these methods. Precisely, the complexity of Sinkhorn method is of order $\mathcal{O}(N^2 \log(N)/\lambda^2)$, where $\lambda$ is the regularization parameter Dvurechensky et al. (2018); Altschuler et al. (2017). Additionally, while entropic regularization interpolates between Maximum-Mean Discrepancy (MMD) (Gretton et al., 2012) and the Wasserstein distance (Feydy et al., 2019), it does not produce a true metric between probability measures. Despite not providing a metric, the entropic OT provides a transport plan, i.e., soft correspondences, albeit not the optimal one. On the other hand, sliced-Wasserstein distances offer linearithmic computational complexity, enabling the comparison of discrete measures with millions of samples. These distances are also topologically equivalent to the Wasserstein distance and offer statistical advantages, such as better sample complexity (Nadjahi et al., 2020). However, despite their computational efficiency, the sliced-Wasserstein approaches do not provide a transport plan between the input probability measures, limiting their applicability to problems that require explicit coupling between measures.

In this paper, we address two central questions: First, **can a transportation plan be constructed between two probability measures using the sliced transport framework?** If so, **can the resulting transportation plan be used to define a metric between the two probability measures?** Within the sliced transport framework, the "slices" refer to the one-dimensional marginals of the source and target probability measures, for which an optimal transportation plan is computed. Crucially, this optimal transportation plan applies to the marginals (i.e., one-dimensional probability measures) rather than the original measures. To derive a transportation plan between the source and target measures, this optimal plan for the marginals must be "lifted" back to the original space.

For discrete measures with equal support size, $N$, and uniform mass, $1/N$, the optimal transportation plan between marginals is represented by a correspondence matrix, specifically an $N \times N$ permutation matrix. Previous works have used directly the correspondence matrix obtained for a slice as a transportation plan in the original space of measures (Rowland et al., 2019; Mahey et al., 2023). This paper provides a holistic and rigorous analysis of this problem for general discrete probability measures.

Our specific contributions in this paper are:

1. Introducing a computationally efficient transport plan between discrete probability measures, the **Expected Sliced Transport (EST) plan**. Motivated by the first question highlighted above, we construct this transport plan as the average of transport plans computed via a lifting scheme involving one-dimensional sliced transport plans. (See Definition 2.4 below.)

2. Providing a distance for discrete probability measures, the **Expected Sliced Transport (EST) distance**. (See Definition 2.6 and Theorem 2.10 below.)

3. Offering both a theoretical proof and an experimental visualization showing that the EST distance is equivalent to the Wasserstein distance (and to weak* convergence) when applied to discrete measures.

4. Illustrating the potential applicability of the proposed distance and transport plan, with a focus on interpolation and classification tasks.

## 2 EXPECTED SLICED TRANSPORT

### 2.1 PRELIMINARIES

Given a probability measure $\mu \in \mathcal{P}(\mathbb{R}^d)$ and a unit vector $\theta \in \mathbb{S}^{d-1} \subset \mathbb{R}^d$, we define $\theta_{\#}\mu := \langle \theta, \cdot \rangle_{\#}\mu$ to be the $\theta$-*slice* of the measure $\mu$, where $\langle \theta, x \rangle = \theta \cdot x = \theta^T x$ denotes the standard inner product in $\mathbb{R}^d$. For any pair of probability measures with finite $p$-moment ($p > 1$) $\mu^1, \mu^2 \in \mathcal{P}_p(\mathbb{R}^d)$, one can pose the following two Optimal Transport (OT) problems: On the one hand, consider the classical OT problem, which gives rise to the $p$-Wasserstein metric:

$$W_p(\mu^1, \mu^2) := \min_{\gamma \in \Gamma(\mu^1, \mu^2)} \left( \int_{\mathbb{R}^d \times \mathbb{R}^d} \|x - y\|^p d\gamma(x, y) \right)^{1/p} \tag{1}$$

Figure 1: Our goal in this paper is to construct transportation plans between $d$-dimensional measures ($d > 1$) using their one-dimensional marginals, or slices. For two measures $\mu^1$ (represented by green circles) and $\mu^2$ (represented by blue circles), we first derive a one-dimensional transport plan $\Lambda_\theta^{\mu^1,\mu^2}$ from their slices along a unit vector $\theta$ (indicated by triangles). This one-dimensional transport plan is then lifted back to the original $d$-dimensional space to produce a transport plan between the measures, $\gamma_\theta^{\mu^1,\mu^2}$. Finally, these transport plans are aggregated over all directions $\theta \in \mathbb{S}^{d-1}$. In (a), the measures $\mu^1$ and $\mu^2$ are uniform, with their masses non-overlapping when projected in the direction of $\theta$. In (b), $\mu^1$ and $\mu^2$ are non-uniform, resulting in overlapping masses when projected along $\theta$. For further details, see Remark A.1 in Appendix A.1.

where $\| \cdot \|$ denotes the Euclidean norm in $\mathbb{R}^d$ and $\Gamma(\mu^1, \mu^2) \subset \mathcal{P}(\mathbb{R}^d \times \mathbb{R}^d)$ is the subset of all probability measures with marginals $\mu^1$ and $\mu^2$. On the other hand, for a given $\theta \in \mathbb{S}^{d-1}$, consider the one-dimensional OT problem:

$$W_p(\theta_\#\mu^1, \theta_\#\mu^2) = \min_{\Lambda_\theta \in \Gamma(\theta_\#\mu^1, \theta_\#\mu^2)} \left( \int_{\mathbb{R}\times\mathbb{R}} |u - v|^p d\Lambda_\theta(u,v) \right)^{1/p} \tag{2}$$

In this case, since the measures $\theta_\#\mu^1, \theta_\#\mu^2$ can be regarded as one-dimensional probabilities in $\mathcal{P}(\mathbb{R})$, there exists a unique optimal transport plan, which we denote by $\Lambda_\theta^{\mu_1,\mu_2}$ (see, for e.g., (Villani, 2021, Thm. 2.18, Remark 2.19)], (Maggi, 2023, Thm. 16.1)).

## 2.2 ON SLICING AND LIFTING TRANSPORT PLANS

In this section, given discrete probability measures $\mu^1, \mu^2 \in \mathcal{P}_p(\mathbb{R}^d)$, we describe the process of slicing them according to a direction $\theta \in \mathbb{S}^{d-1}$ and lifting the optimal transport plan $\Lambda_\theta^{\mu^1,\mu^2}$, which solves the 1-d OT problem (2), to get a plan in $\Gamma(\mu^1, \mu^2)$. Thus, we obtain a new measure, denoted as $\gamma_\theta^{\mu^1,\mu^2}$, in $\mathcal{P}(\mathbb{R}^d \times \mathbb{R}^d)$ with first and second marginal $\mu^1$ and $\mu^2$, respectively. For clarity, we first describe the process for discrete uniform measures and then extend it to any pair of discrete measures.

### 2.2.1 ON SLICING AND LIFTING TRANSPORT PLANS FOR **UNIFORM DISCRETE MEASURES**

Given $N \in \mathbb{N}$, consider the space $\mathcal{P}_{(N)}(\mathbb{R}^d)$ of uniform discrete probability measures concentrated at $N$ particles in $\mathbb{R}^d$, that is,

$$\mathcal{P}_{(N)}(\mathbb{R}^d) = \left\{ \frac{1}{N} \sum_{i=1}^N \delta_{x_i} \mid x_i \in \mathbb{R}^d, \ \forall i \in \{1, ..., N\} \right\}.$$

Let $\mu^1 = \frac{1}{N} \sum_{i=1}^N \delta_{x_i}, \mu^2 = \frac{1}{N} \sum_{j=1}^N \delta_{y_j} \in \mathcal{P}_{(N)}(\mathbb{R}^d)$, where $x_i, y_j \in \mathbb{R}^d$ and $\delta_{x_i}$ denotes a Dirac measure located at $x_i$ (respectively for $\delta_{y_j}$). Let us denote by $\mathcal{U}(\mathbb{S}^{d-1})$ the uniform measure on the hypersphere $\mathbb{S}^{d-1} \subset \mathbb{R}^d$. In this case, the $\theta$-slice of $\mu^1$ is represented by $\theta_\#\mu^1 = \frac{1}{N} \sum_{i=1}^N \delta_{\theta \cdot x_i}$, and similarly for $\theta_\#\mu^2$. Let $\mathbf{S}_N$ denote the symmetric group of all permutations of the elements in the set $[N] := \{1, \ldots, N\}$. Let $\zeta_\theta, \tau_\theta \in \mathbf{S}_N$ denote the sorted indices of the projected points $\{\theta \cdot x_i\}_{i=1}^N$ and $\{\theta \cdot y_j\}_{j=1}^N$, respectively, that is,

$$\theta \cdot x_{\zeta_\theta^{-1}(1)} \leq \theta \cdot x_{\zeta_\theta^{-1}(2)} \leq \cdots \leq \theta \cdot x_{\zeta_\theta^{-1}(N)} \text{ and } \theta \cdot y_{\tau_\theta^{-1}(1)} \leq \theta \cdot y_{\tau_\theta^{-1}(2)} \leq \cdots \leq \theta \cdot y_{\tau_\theta^{-1}(N)} \tag{3}$$

The optimal matching from $\theta_\#\mu^1$ to $\theta_\#\mu^2$ for the problem (2) is induced by the assignment

$$\theta \cdot x_{\zeta_\theta^{-1}(i)} \longmapsto \theta \cdot y_{\tau_\theta^{-1}(i)}, \qquad \forall 1 \leq i \leq N. \tag{4}$$

We define $T_\theta^{\mu^1,\mu^2} : \{x_1, \ldots, x_N\} \to \{y_1, \ldots, y_N\}$ the *lifted transport map* between $\mu^1$ and $\mu^2$ by:

$$T_\theta^{\mu^1,\mu^2}(x_i) = y_{\tau_\theta^{-1}(\zeta_\theta(i))}, \qquad \forall 1 \leq i \leq N. \tag{5}$$

Rigorously, $T_\theta^{\mu^1,\mu^2}$ is not necessarily a function defined on $\{x_1, \ldots, x_N\}$ but on the labels $\{1, \ldots, N\}$, as two projected points $\theta \cdot x_i$ and $\theta \cdot x_j$ could coincide for $i \neq j$. As a result, it is more appropriate to work with *lifted transport plans*. Indeed, the matrix $u_\theta^{\mu^1,\mu^2} \in \mathbb{R}^{n \times n}$ given by

$$u_\theta^{\mu^1,\mu^2}(i,j) = \begin{cases} 1/N & \text{if } j = \tau_\theta^{-1}(\zeta_\theta(i)) \\ 0 & \text{otherwise} \end{cases} \tag{6}$$

encodes the weights of the optimal transport plan between $\theta_\#\mu^1$ and $\theta_\#\mu^2$ given by

$$\Lambda_\theta^{\mu^1,\mu^2} := \sum_{i,j} u_\theta^{\mu^1,\mu^2}(i,j)\delta_{(\theta \cdot x_i, \theta \cdot y_j)}, \tag{7}$$

as well as the weights of the *lifted transport plan* between the original measures $\mu^1$ and $\mu^2$ according to the $\theta$-slice defined by

$$\gamma_\theta^{\mu^1,\mu^2} := \sum_{i,j} u_\theta^{\mu^1,\mu^2}(i,j)\delta_{(x_i,y_j)}. \tag{8}$$

This new measure $\gamma_\theta^{\mu^1,\mu^2} \in \mathcal{P}(\mathbb{R}^d \times \mathbb{R}^d)$ has marginals $\mu^1$ and $\mu^2$. While $\gamma_\theta^{\mu^1,\mu^2}$ is not necessarily optimal for problem (1), it can be interpreted as a transport plan in $\Gamma(\mu^1, \mu^2)$ which is optimal when projecting $\mu^1$ and $\mu^2$ in the direction of $\theta$. See Figure 1 (a) for a visualization.

### 2.2.2 ON SLICING AND LIFTING TRANSPORT PLANS FOR GENERAL DISCRETE MEASURES

Consider discrete measures $\mu^1, \mu^2 \in \mathcal{P}_p(\mathbb{R}^d)$. In this section, we will use the notation $\mu^1 = \sum_{x \in \mathbb{R}^d} p(x)\delta_x$, where $p(x) \geq 0$ for all $x \in \mathbb{R}^d$, $p(x) \neq 0$ for at most countable many points $x \in \mathbb{R}^d$, and $\sum_{x \in \mathbb{R}^d} p(x) = 1$. Similarly, $\mu^2 = \sum_{y \in \mathbb{R}^d} q(y)\delta_y$ for a non-negative density function $q$ in $\mathbb{R}^d$ with finite or countable support and such that $\sum_{y \in \mathbb{R}^d} q(y) = 1$. Given $\theta \in \mathbb{S}^{d-1}$, consider the equivalence relation defined by:

$$x \sim_\theta x' \quad \text{if and only if} \quad \theta \cdot x = \theta \cdot x'$$

We denote by $\bar{x}^\theta$ the equivalence class of $x \in \mathbb{R}^d$. By abuse of notation, we will use interchangeably that $\bar{x}^\theta$ is a point in the quotient space $\mathbb{R}^d/\sim_\theta$, and also the set $\{x' \in \mathbb{R}^d : \theta \cdot x = \theta \cdot x'\}$, which is the orthogonal hyperplane to $\theta$ that intersects $x$. The intended meaning of $\bar{x}^\theta$ will be clear from the context. Notice that, geometrically, the quotient space $\mathbb{R}^d/\sim_\theta$ is the line $\mathbb{R}$ in the direction of $\theta$.

Now, we interpret the projected measures $\theta_\#\mu^1$, $\theta_\#\mu^2$ as 1-dimensional probability measures in $\mathcal{P}(\mathbb{R}^d/\sim_\theta)$ given by $\theta_\#\mu^1 = \sum_{\bar{x}^\theta \in \mathbb{R}^d/\sim_\theta} P(\bar{x}^\theta)\delta_{\bar{x}^\theta}$, where $P(\bar{x}^\theta) = \sum_{x' \in \bar{x}^\theta} p(x')$, and similarly, $\theta_\#\mu^2 = \sum_{\bar{y}^\theta \in \mathbb{R}^d/\sim_\theta} Q(\bar{y}^\theta)\delta_{\bar{y}^\theta}$, where $Q(\bar{y}^\theta) = \sum_{y' \in \bar{y}^\theta} q(y')$.

**Remark 2.1.** *Notice that if $P(\bar{x}^\theta) = 0$, then $p(x') = 0$ for all $x' \in \bar{x}^\theta$, or, equivalently, if $p(x) \neq 0$, then $P(\bar{x}^\theta) \neq 0$ (where $x$ is any 'representative' of the class $\bar{x}^\theta$). Similarly for $Q$.*

Consider the optimal transport plan $\Lambda_\theta^{\mu^1,\mu^2} \in \Gamma(\theta_\#\mu^1, \theta_\#\mu^2) \subset \mathcal{P}(\mathbb{R}^d/\sim_\theta \times \mathbb{R}^d/\sim_\theta)$ between $\theta_\#\mu^1$ and $\theta_\#\mu^2$, which is *unique* for the OT problem (2) as we are considering 1-dimensional probability measures. Let us define

$$u_\theta^{\mu^1,\mu^2}(x,y) := \begin{cases} \frac{p(x)q(y)}{P(\bar{x}^\theta)Q(\bar{y}^\theta)} \Lambda_\theta^{\mu^1,\mu^2}(\{(\bar{x}^\theta, \bar{y}^\theta)\}) & \text{if } p(x) \neq 0 \text{ and } q(y) \neq 0 \\ 0 & \text{if } p(x) = 0 \text{ or } q(y) = 0 \end{cases}$$

which allows us to generalize the *lifted transport plan* given in (6) in the general discrete case:

$$\gamma_\theta^{\mu^1,\mu^2} := \sum_{x \in \mathbb{R}^d} \sum_{y \in \mathbb{R}^d} u_\theta^{\mu^1,\mu^2}(x,y)\delta_{(x,y)} \tag{9}$$

See Figure 1 (b) for a visualization.

**Remark 2.2.** *Notice that this lifting process can be performed by starting with any transport plan $\Lambda_\theta \in \Gamma(\theta_\#\mu^1, \theta_\#\mu^2)$, but in this article we will always consider the optimal transport plan, i.e., $\Lambda_\theta = \Lambda_\theta^{\mu^1,\mu^2}$. The reason why we make this choice is because this will give rise to a metric between discrete probability measures: The EST distance which will be defined in Section 2.3.*

**Lemma 2.3.** *Given general discrete probability measures $\mu^1$ and $\mu^2$ in $\mathbb{R}^d$, the discrete measure $\gamma_\theta^{\mu^1,\mu^2}$ defined by (9) has marginals $\mu^1$ and $\mu^2$, that is, $\gamma_\theta^{\mu^1,\mu^2} \in \Gamma(\mu^1, \mu^2) \subset \mathcal{P}(\mathbb{R}^d \times \mathbb{R}^d)$.*

We refer the reader to the appendix for its proof.

## 2.3 EXPECTED SLICED TRANSPORT (EST) FOR DISCRETE MEASURES

Leveraging on the transport plans $\gamma_\theta^{\mu^1,\mu^2}$ described before, in this section we propose a new transport plan $\bar{\gamma}^{\mu^1,\mu^2} \in \Gamma(\mu^1,\mu^2)$, which will give rise to a new metric in the space of discrete probability measures.

**Definition 2.4** (Expected Sliced Transport plan). *Let $\sigma \in \mathcal{P}(\mathbb{S}^{d-1})$. For discrete probability measures $\mu^1,\mu^2$ in $\mathbb{R}^d$, we define the **Expected Sliced Transport (EST) plan** $\bar{\gamma}^{\mu^1,\mu^2} \in \mathcal{P}(\mathbb{R}^d \times \mathbb{R}^d)$ by*

$$\bar{\gamma}^{\mu^1,\mu^2} := \mathbb{E}_{\theta \sim \sigma}[\gamma_\theta^{\mu^1,\mu^2}], \qquad \text{where each } \gamma_\theta^{\mu^1,\mu^2} \text{ is given by (9), that is,} \tag{10}$$

$$\bar{\gamma}^{\mu^1,\mu^2}(\{(x,y)\}) = \int_{\mathbb{S}^{d-1}} \gamma_\theta^{\mu^1,\mu^2}(\{(x,y)\})d\sigma(\theta), \qquad \forall x,y \in \mathbb{R}^d \times \mathbb{R}^d.$$

*In other words, $\bar{\gamma}^{\mu^1,\mu^2} = \sum_{x \in \mathbb{R}^d} \sum_{y \in \mathbb{R}^d} U^{\mu^1,\mu^2}(x,y)\delta_{(x,y)}$, where the new weights are given by*

$$U^{\mu^1,\mu^2}(x,y) = \begin{cases} p(x)q(y) \int_{\mathbb{S}^{d-1}} \frac{\Lambda_\theta^{\mu^1,\mu^2}(\{(\bar{x}^\theta,\bar{y}^\theta)\})}{P(\bar{x}^\theta)Q(\bar{y}^\theta)}d\sigma(\theta) & \text{if } p(x) \neq 0 \text{ and } q(y) \neq 0 \\ 0 & \text{otherwise} \end{cases}$$

**Remark 2.5.** *The measure $\bar{\gamma}^{\mu^1,\mu^2}$ is well-defined and, moreover, (as an easy consequence of Lemma 2.3) $\bar{\gamma}^{\mu^1,\mu^2} \in \Gamma(\mu^1,\mu^2)$, i.e., it has marginals $\mu^1$ and $\mu^2$. (See Lemma A.5 in the appendix.)*

**Definition 2.6** (Expected Sliced Transport distance). *Let $\sigma \in \mathcal{P}(\mathbb{S}^{d-1})$ with $\text{supp}(\sigma) = \mathbb{S}^{d-1}$. We define the **Expected Sliced Transport discrepancy** for discrete probability measures $\mu^1,\mu^2$ in $\mathbb{R}^d$ by*

$$\mathcal{D}_p(\mu^1,\mu^2) := \left( \sum_{x \in \mathbb{R}^d} \sum_{y \in \mathbb{R}^d} \|x-y\|^p \, \bar{\gamma}^{\mu^1,\mu^2}(\{(x,y)\}) \right)^{\frac{1}{p}}, \text{ where } \bar{\gamma}^{\mu^1,\mu^2} \text{ is defined by (10).} \tag{11}$$

**Remark 2.7.** *By defining the following generalization of the Sliced Wasserstein Generalized Geodesics (SWGG) dissimilarity presented in Mahey et al. (2023),*

$$\mathcal{D}_p(\mu^1,\mu^2;\theta) := \left( \sum_{x \in \mathbb{R}^d} \sum_{y \in \mathbb{R}^d} \|x-y\|^p \gamma_\theta^{\mu^1,\mu^2}(\{x,y\}) \right)^{1/p}, \tag{12}$$

*we can rewrite (11) as*

$$\mathcal{D}_p(\mu^1,\mu^2) = \mathbb{E}_{\theta \sim \sigma}^{1/p}[\mathcal{D}_p^p(\mu^1,\mu^2;\theta)]$$

**Remark 2.8.** *Since the EST plan $\bar{\gamma}^{\mu^1,\mu^2}$ is a transport plan, we have that*

$$W_p(\mu^1,\mu^2) \leq \mathcal{D}_p(\mu^1,\mu^2).$$

*In Appendix B we show that they define the same topology for the space of discrete probabiliies.*

**Remark 2.9** (EST for discrete uniform measures and the Projected Wasserstein distance). *Consider uniform measures $\mu^1 = \frac{1}{N}\sum_{i=1}^N \delta_{x_i}, \mu^2 = \frac{1}{N}\sum_{j=1}^N \delta_{y_j} \in \mathcal{P}_{(N)}(\mathbb{R}^d)$, and for $\theta \in \mathbb{S}^{d-1}$, let $\zeta_\theta, \tau_\theta \in \mathbf{S}_N$ be permutations that allow us to order the projected points as in (3). Notice that if $\sigma = \mathcal{U}(\mathbb{S}^{d-1})$, by using the formula (5) for each assignment given $\theta$ and noticing that $\tau_\theta^{-1} \circ \zeta_\theta \in \mathbf{S}_N$, we can re-write (11) as*

$$\mathcal{D}_p(\mu^1,\mu^2)^p = \mathbb{E}_{\theta \sim \mathcal{U}(\mathbb{S}^{d-1})}\left[\frac{1}{N}\sum_{i=1}^N \|x_i - y_{\tau_\theta^{-1}(\zeta_\theta(i))}\|^p\right]. \tag{13}$$

*Therefore, we have that the expression for $\mathcal{D}_p(\cdot,\cdot)$ given by (13) coincides with the **Projected Wasserstein distance** proposed in (Rowland et al., 2019, Definition 3.1). Then, by applying (Rowland et al., 2019, Proposition 3.3), we have that the Expected Sliced Transport discrepancy defined in Equation (13) is a metric on the space $\mathcal{P}_{(N)}(\mathbb{R}^d)$. We generalise this in the next theorem.*

**Theorem 2.10.** *The Expected Sliced Transport $\mathcal{D}_p(\cdot,\cdot)$ defined in (11) is a metric in the space of finite discrete probability measures in $\mathbb{R}^d$.*

*Sketch of the proof of Theorem 2.10.* For the detailed proof, we refer the reader to Appendix A. Here, we present a brief overview of the main ideas and steps involved in the proof.

The symmetry of $\mathcal{D}_p(\cdot, \cdot)$ follows from our construction of the transport plan $\bar{\gamma}^{\mu^1, \mu^2}$, which is based on considering a family of *optimal* 1-d transport plans $\{\Lambda_\theta^{\mu^1, \mu^2}\}_{\theta \in \mathbb{S}^{d-1}}$. The identity of indiscernibles follows essentially from Remark 2.8. To prove the triangle inequality we use the following strategy:

1. We leverage the fact that $\mathcal{D}_p(\cdot, \cdot)$ is a metric for the space $\mathcal{P}_{(N)}(\mathbb{R}^d)$ of uniform discrete probability measures concentrated at $N$ particles in $\mathbb{R}^d$ (Rowland et al. (2019)) to prove that $\mathcal{D}_p(\cdot, \cdot)$ is also a metric on the set of measures in which the masses are rationals. To do so, we establish a correspondence between finite discrete measures with rational weights and finite discrete measures with uniform mass (see the last paragraph of Proposition A.7).

2. Given finite discrete measures $\mu^1, \mu^2$, we approximate them, in terms of the Total Variation norm, by sequences of probability measures with rational weights $\{\mu_n^1\}, \{\mu_n^2\}$, supported on the same points as $\mu^1$ and $\mu^2$, respectively. We then turn our attention on how the various plans constructed behave as the $n$ increases and show the following convergence results in Total Variation norm:

   (a) The sequence $\left(\Lambda_\theta^{\mu_n^1, \mu_n^2}\right)_{n \in \mathbb{N}}$ converges to $\Lambda_\theta^{\mu^1, \mu^2}$.
   (b) The sequence $\left(\gamma_\theta^{\mu_n^1, \mu_n^2}\right)_{n \in \mathbb{N}}$ converges to $\gamma_\theta^{\mu^1, \mu^2}$.
   (c) The sequence $\left(\bar{\gamma}^{\mu_n^1, \mu_n^2}\right)_{n \in \mathbb{N}}$ converges to $\bar{\gamma}^{\mu^1, \mu^2}$.

   As a consequence, we obtain $\lim_{n \to \infty} \mathcal{D}_p(\mu_n^1, \mu_n^2) = \mathcal{D}_p(\mu^1, \mu^2)$.

Finally, given three finite discrete measures $\mu^1, \mu^2, \mu^3$, we proceed as in point 2 by considering sequences of probability measures with rational weights $\{\mu_n^1\}, \{\mu_n^2\}, \{\mu_n^3\}$ supported on the same points as $\mu^1, \mu^2, \mu^3$, respectively, that approximate the original measures in Total Variation, obtaining

$$\mathcal{D}_p(\mu^1, \mu_n^2) = \lim_{n \to \infty} \mathcal{D}_p(\mu_n^1, \mu_n^2) \leq \lim_{n \to \infty} \mathcal{D}_p(\mu_n^1, \mu_n^3) + \mathcal{D}_p(\mu_n^3, \mu_n^2) = \mathcal{D}_p(\mu^1, \mu^3) + \mathcal{D}_p(\mu^3, \mu^2)$$

where the equalities follows from point 2 and the middle triangle inequality follows from point 1. $\square$

## 3 EXPERIMENTS

### 3.1 COMPUTATIONAL EFFICIENCY

In practice, to compute the EST plan (10) and distance (11), we sample $L$ unit vectors $\{\theta_\ell\}_{\ell=1}^L$ (or *slices*) uniformly from $\mathbb{S}^{d-1}$ to approximate $\bar{\gamma}^{\mu^1, \mu^2} \approx \frac{1}{L} \sum_{\ell=1}^L \gamma_{\theta_\ell}^{\mu^1, \mu^2}$. For empirical measures $\mu^1, \mu^2$ in $\mathbb{R}^d$ of size $N$, the computational complexity of our proposed EST method is of order $\mathcal{O}((L+d)N^2)$. A detailed analysis is provided in Appendix C. Additionally, Figure 2 presents the wall-clock time of the Sinkhorn al-

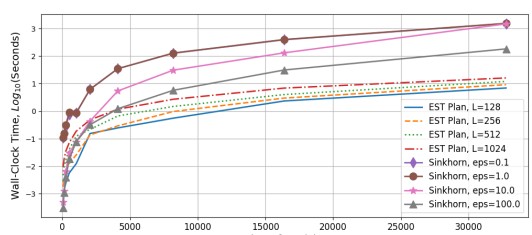

Figure 2: Wall-clock time plot of Sinkhorn algorithm for entropic OT with varying $\lambda$ and the EST method for different numbers of slices, $L$, as the size $N$ of the empirical measures increases.

gorithm, used to compute entropic OT for various values of the regularization parameter $\lambda$, and the proposed EST method, evaluated with different numbers $L$ of slices, as the size $N$ of the support of the discrete measures increases. The wall clock times are averaged over 10 runs.

### 3.2 COMPARISON OF TRANSPORT PLANS

Figure 3 illustrates the behavior of different transport schemes: the optimal transport plan for $W_2(\cdot, \cdot)$, the transport plan obtained by solving an entropically regularized transportation problem between the source and target probability measures, and the new expected sliced transport plan $\bar{\gamma}$. We include comparisons with entropic regularization because it is one of the most popular approaches, as it allows for the use of Sinkhorn's algorithm. From the figure, we observe that while $\bar{\gamma}$ promotes mass splitting, this phenomenon is less pronounced than in the entropically regularized OT scheme. This observation will be revisited in Subsection 3.3.

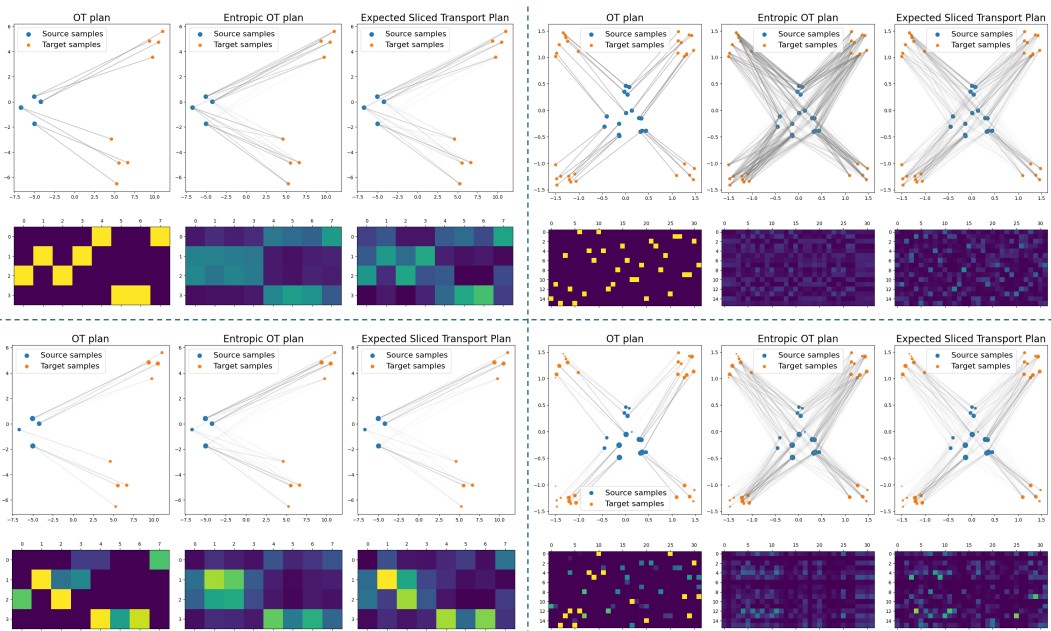

Figure 3: Depiction of transport plans (an optimal transport plan, a plan obtained from solving an entropically regularized transport problem, and the proposed expected sliced transport plan) between source (orange) and target (blue) for four different configurations of masses. The measures in the left and right panels are concentrated on the same particles, respectively; however, the top row depicts measures with uniform mass, while the bottom row depicts measures with random, non-uniform mass. transport plans are shown as gray assignments and as $n \times m$ heat matrices encoding the amount of mass transported (dark color = no transportation, bright color = more transportation), where $n$ is the number of particles on which the source measure is concentrated, and $m = 2n$) is the number of particles on which the target measure is concentrated.

### 3.3 TEMPERATURE APPROACH

Given $\mu^1, \mu^2$ discrete probability measures, we perform the new expected sliced transportation scheme by using the following averaging measure $\sigma_\tau \ll \mathcal{U}(\mathbb{S}^{d-1})$ on the sphere:

$$d\sigma_\tau(\theta) = \frac{e^{-\tau \mathcal{D}_p^p(\mu^1,\mu^2;\theta)}}{\int_{\mathbb{S}^{d-1}} e^{-\tau \mathcal{D}_p^p(\mu^1,\mu^2;\theta')} d\theta'} d\theta, \tag{14}$$

where $\mathcal{D}_p(\mu^1, \mu^2; \theta)$ is given by (12), and $\tau \geq 0$ is a hyperparameter we will refer to as the *temperature* (note that increasing $\tau$ corresponds to reducing the temperature). If $\tau = 0$, then $\sigma_0 = \mathcal{U}(\mathbb{S}^{d-1})$. However, when $\tau \neq 0$, $\sigma_\tau$ is a probability measure on $\mathbb{S}^{d-1}$ with density given by (14), which depends on the source and target measures $\mu^1$ and $\mu^2$. We have chosen this measure $\sigma_\tau$ because it provides a general parametric framework that interpolates between our proposed scheme with the uniform measure ($\tau = 0$) and min-SWGG (Mahey et al., 2023), as the EST distance approaches min-SWGG as $\tau \to \infty$. For the implementations, we use

$$\sigma_\tau(\theta^l) = \frac{e^{-\tau \mathcal{D}_p^p(\mu^1,\mu^2;\theta^l)}}{\sum_{\ell'=1}^{L} e^{-\tau \mathcal{D}_p^p(\mu^1,\mu^2;\theta^{\ell'})}}, \tag{15}$$

where $L$ represents the number of slices or unit vectors $\theta^1, \ldots, \theta^L \in \mathbb{S}^{d-1}$. Figure 4 illustrates that as $\tau \to \infty$, the weights used for averaging the lifted transport plans converge to a one-hot vector, i.e., the slice minimizing $\mathcal{D}_p(\mu^1, \mu^2; \theta)$ dominates, leading to a transport plan with fewer mass splits. For the visualization we have used source $\mu^1$ and target $\mu^2$ uniform probability measures concentrated on different number of particles. For consistency, the configurations are the same as in Figure 3.

### 3.4 INTERPOLATION

We use the Point Cloud MNIST 2D dataset Garcia (2023), a reimagined version of the classic MNIST dataset (LeCun, 1998), where each image is represented as a set of weighted 2D point clouds instead

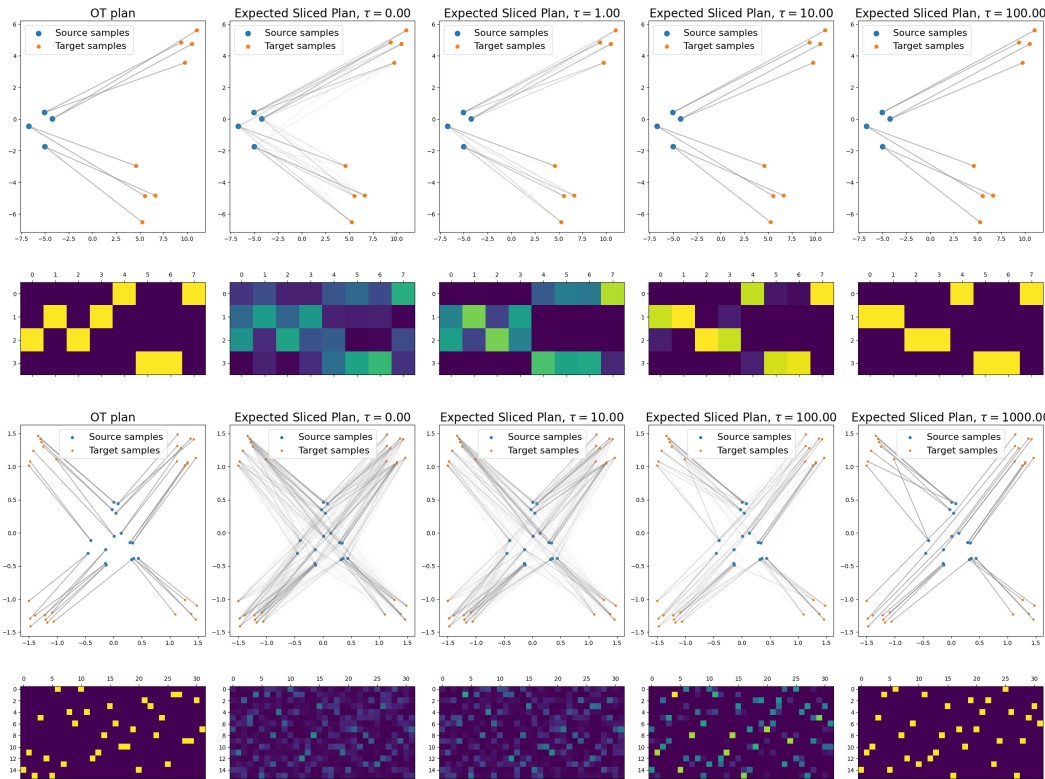

Figure 4: The effect of increasing $\tau$ (i.e., decreasing temperature) on the expected sliced plan. The left most column shows the OT plan, and the rest of the columns show the expected sliced plan as a function of increasing $\tau$. The right most column depicts that expected sliced plan recovers the min-SWGG Mahey et al. (2023) transportation map.

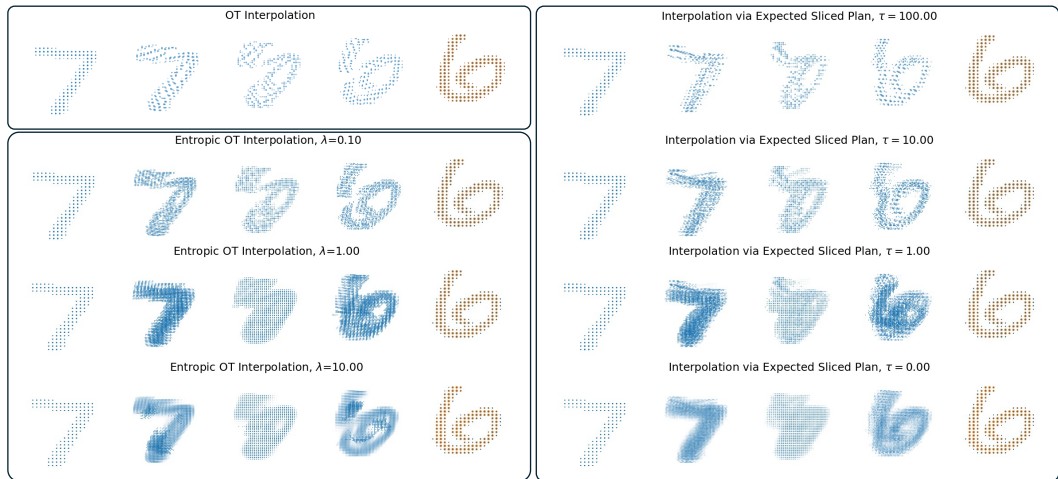

Figure 5: Interpolation between two point clouds via $((1-t)x + ty)_{\#}\gamma$, where $\gamma$ is the optimal transport plan for $W_2(\cdot, \cdot)$ (top left), the transport plan obtained from entropic OT with various regularization parameters (bottom left), and the EST for different temperatures $\tau$ (right).

of pixel values. In Figure 5, we illustrate the interpolation between two point clouds that represent digits 7 and 6. Since the point clouds are discrete probability measures with non-uniform mass, we perform three different interpolation schemes via $((1-t)x + ty)_{\#}\gamma$ where $0 \leq t \leq 1$ for different transport plans $\gamma$, namely:

1. $\gamma = \gamma^*$, an optimal transport plan for $W_2(\cdot, \cdot)$;

2. a transport plan $\gamma$ obtained from solving an entropically regularized transportation problem (performed for three different regularization parameters $\lambda$);

3. $\gamma = \bar{\gamma}$: the expected sliced transport plan computed using $\sigma_\tau$ given by formula (14) (or (15) for implementations) for four different values of the temperature parameter $\tau$.

As the temperature increases, the transport plan exhibits less mass splitting, similar to the effect of decreasing the regularization parameter $\lambda$ in entropic OT. However, unlike entropic OT, where smaller regularization parameters require more iterations for convergence, the computation time for expected sliced transport remains unaffected by changes in temperature.

### 3.5 WEAK CONVERGENCE

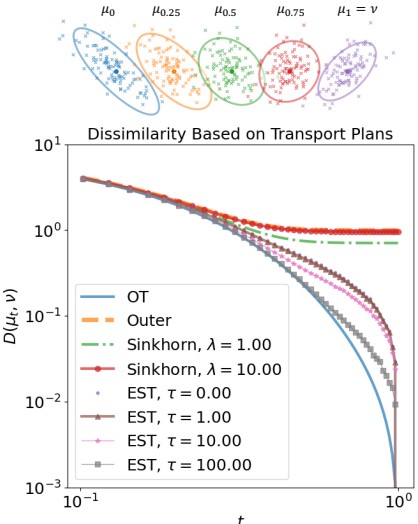

Given finite discrete probability measures $\mu$ and $\nu$, we consider $\mu_t$, for $0 \le t \le 1$, the Wasserstein geodesic between $\mu$ and $\nu$. In particular, $\mu_t$ is a curve of probability measures that interpolates $\mu$ and $\nu$, that is $\mu_0 = \mu$ and $\mu_1 = \nu$. Moreover, we have that $W_2(\mu_t, \nu) = (1 - t)W_2(\mu, \nu) \longrightarrow 0$ as $t \to 1$, or equivalently, we can say $\mu_t$ converges in the weak*-topology to $\nu$. Figure 6 illustrates that the expected sliced distance also satisfies $\mathcal{D}_2(\mu_t, \nu) \longrightarrow 0$ as $t \to 1$. Indeed, this experimental conclusion is justified by the following theoretical result:

*Let $\mu, \mu_n \in \mathcal{P}(\Omega)$ be discrete measures with finite or countable support, where $\Omega \subset \mathbb{R}^d$ is compact. Assume $\sigma \ll \mathcal{U}(\mathbb{S}^{d-1})$. Then, $\mathcal{D}_p(\mu_n, \mu) \to 0$ if and only if $\mu_n \overset{*}{\rightharpoonup} \mu$.*

We present its proof in Appendix B.

Figure 6: Discrepancies calculated from transport plans between $\mu_t$ and $\nu$, when $\mu_t \overset{*}{\rightharpoonup} \nu$, as a function of $t \in [0, 1]$.

For the experiment, $\mu$ and $\nu$ are chosen to be discrete measures with $N$ particles of uniform mass, sampled from two Gaussian distributions (see Figure 6, top). For different values of time, $0 \le t \le 1$, we compute different discrepancies, $\sum_{i=1}^{N} \sum_{j=1}^{N} \|x_i - y_j\|^2 \gamma_{ij}^{\mu_t, \nu}$, calculated for various transport plans: (1) the optimal transport plan, (2) the outer product plan $\mu_t \otimes \nu$, (3) the plan obtained from entropic OT with two different regularization parameters $\lambda$, and (4) our proposed expected sliced plan computed with $\sigma_\tau$ given in (14) for four different temperature parameters $\tau$. As $\mu_t$ converges to $\nu$, it is evident that both the OT and our proposed EST distance approach zero, while the entropic OT and outer product plans, as expected, do not converge to zero.

### 3.6 TRANSPORT-BASED EMBEDDING

Following the linear optimal transportation (LOT) framework, also referred to as the Wasserstein or transport-based embedding framework (Wang et al., 2013; Kolouri et al., 2021; Nenna & Pass, 2023; Bai et al., 2023; Martín et al., 2024), we investigate the application of our proposed transport plan in point cloud classification. Let $\mu_0 = \sum_{i=1}^{N} \alpha_i \delta_{x_i}$ denote a reference probability measure and let $\mu_k = \sum_{j=1}^{N_k} \beta_j^k \delta_{y_j^k}$ denote a target probability measure. Let $\gamma^{\mu_0, \mu_k}$ be a transport plan between $\mu_0$ and $\mu_k$, and define the barycentric projection (Ambrosio et al., 2011, Definition 5.4.2) of this plan as:

$$b_i(\gamma^{\mu_0, \mu_k}) := \frac{1}{\alpha_i} \sum_{j=1}^{N_k} \gamma_{ij}^{\mu_0, \mu_k} y_j^k, \qquad i \in 1, ..., N. \tag{16}$$

Note that $b_i(\gamma^{\mu_0, \mu_k})$ represents the center of mass to which $x_i$ from the reference measure is transported according to the transport plan $\gamma^{\mu_0, \mu_k}$. When $\gamma^{\mu_0, \mu_k}$ is the OT plan, the LOT framework of Wang et al. (2013) uses

$$[\phi(\mu_k)]_i := b_i(\gamma^{\mu_0, \mu_k}) - x_i, \qquad i \in 1, ..., N \tag{17}$$

as an embedding $\phi$ for the measure $\mu_k$. This framework, as demonstrated in Kolouri et al. (2021), can be used to define a permutation-invariant embedding for sets of features and, more broadly, point clouds. More precisely, given a point cloud $\mathcal{Y}_k = \{(\beta_j^k, y_j^k)\}_{j=1}^{N_k}$, where $\sum_{j=1}^{N_k} \beta_j^k = 1$ and $\beta_j$ represent the mass at location $y_j$, we represent this point cloud as a discrete measure $\mu_k$.

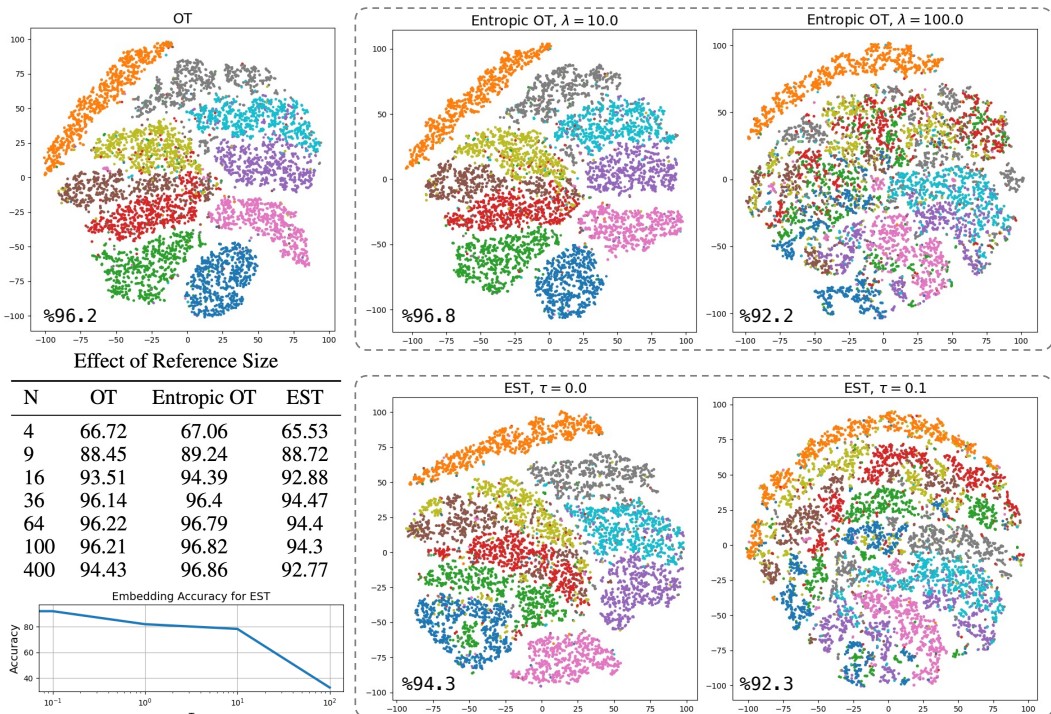

Figure 7: t-SNE visualization of the embeddings computed using different transport plans, along with the corresponding logistic regression accuracy for each embedding. The t-SNE plots are generated for embeddings with a reference size of $N = 100$, and for EST, we used $L = 128$ slices. The table shows the accuracy of the embeddings as a function of reference size $N$. For the table, the regularization parameter for entropic OT is set to $\lambda = 10$, and for EST, the temperature is set to $\tau = 0$ with $L = 128$ slices. Lastly, the plot on the bottom left shows the performance of EST, when $N = 100$ and $L = 128$, as a function of the temperature parameter, $\tau$.

In this section, we use a reference measure with $N$ particles of uniform mass to embed the digits from the Point Cloud MNIST 2D dataset using various transport plans. We then perform a logistic regression on the embedded digits and present the results in Figure 7. The figure shows a 2D t-SNE visualization of the embedded point clouds using: (1) the OT plan, (2) the entropic OT plan with two different regularization parameters, and (3) our expected sliced plan with two temperature parameters (using $N = 100$ for all methods). In addition, we report the test accuracy of these embeddings for different reference sizes.

Lastly, we make an interesting observation about the embedding computed using EST. As we reduce the temperature, i.e., increase $\tau$, the embedding becomes progressively less informative. We attribute this to the dependence of $\sigma_\tau$ on $\mu_k$. In other words, the embedding is computed with respect to different $\sigma_\tau$ for different measures, leading to inaccuracies when comparing the embedded measures. This finding also suggests that the min-SWGG framework, while meritorious, may not be well-suited for defining a transport-based embedding.

Additionally, we refer the reader to Appendix D for a similar classification experiment using a more complex dataset: ModelNet40.

## 4   CONCLUSIONS

In this paper, we explored the feasibility of constructing transport plans between two probability measures using the computationally efficient sliced optimal transport (OT) framework. We introduced the Expected Sliced Transport (EST) framework and proved that it provides a valid metric for comparing discrete probability measures while preserving the computational efficiency of sliced transport and enabling explicit mass coupling. Through a diverse set of numerical experiments, we illustrated the behavior of this newly introduced transport plan. Additionally, we demonstrated how the temperature parameter in our approach offers a flexible framework that connects our method to the recently proposed min-Sliced Wasserstein Generalized Geodesics (min-SWGG) framework. Finally, the theoretical insights and experimental results presented here open up new avenues for developing efficient transport-based algorithms in machine learning and beyond.

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

# A  PROOF OF THEOREM 2.10: METRIC PROPERTY OF THE EXPECTED SLICED DISCREPANCY FOR DISCRETE PROBABILITY MEASURES

## A.1  PRELIMINARIES ON EXPECTED SLICED TRANSPORTATION

**Remark A.1** (Figure 1). *Let us elaborate on explaining Figure 1. (a) Visualization for uniform discrete measures $\mu^1 = \frac{1}{3}(\delta_{x_1} + \delta_{x_2} + \delta_{x_3})$ (green circles), $\mu^2 = \frac{1}{3}(\delta_{y_1} + \delta_{y_2} + \delta_{y_3})$ (blue circles) in $\mathcal{P}_{(N)}(\mathbb{R}^d)$ with $n = 3$. Given an angle $\theta$ (red unit vector), when sorting $\{\theta \cdot x_i\}_{i=1}^3$ and $\{\theta \cdot y_j\}_{j=1}^3$ we use permutations $\zeta_\theta$ and $\tau_\theta$ given by $\zeta_\theta(2) = 1, \zeta_\theta(1) = 2, \zeta_\theta(3) = 3$, and $\tau_\theta(1) = 1, \tau_\theta(3) = 2, \tau_\theta(2) = 3$. The optimal transport map between $\theta_{\#}\mu^1$ (green triangles) and $\theta_{\#}\mu^2$ (blue triangles) is given by the following assignment: $\theta \cdot x_{\zeta_\theta^{-1}(1)} = \theta \cdot x_2 \longmapsto \theta \cdot y_1 = \theta \cdot y_{\tau_\theta^{-1}(1)}$, $\theta \cdot x_{\zeta_\theta^{-1}(2)} = \theta \cdot x_1 \longmapsto \theta \cdot y_3 = \theta \cdot y_{\tau_\theta^{-1}(2)}$, $\theta \cdot x_{\zeta_\theta^{-1}(3)} = \theta \cdot x_3 \longmapsto \theta \cdot y_2 = \theta \cdot y_{\tau_\theta^{-1}(3)}$. This gives rise to the plan $\Lambda_\theta^{\mu^1,\mu^2}$ given in (7) with is represented by solid arrows in the first panel. The lifted plan $\gamma_\theta^{\mu^1,\mu^2}$ defined in (8) is represented in the second panel by dashed assignments. (b) Visualization for finite discrete measures $\mu^1 = 0.1\delta_{x_1} + 0.3\delta_{x_2} + 0.6\delta_{x_3}$ (green circles), $\mu^2 = 0.5\delta_{y_1} + 0.3\delta_{y_2} + 0.2\delta_{y_3}$ (blue circles). When projection according a given direction $\theta$, the locations with green masses $0.3$ and $0.6$ overlap, as well as the locations with blue masses $0.2$ and $0.3$. Thus, the mass of $\theta_{\#}\mu^1$ is concentrated at two green points (triangles) on the line determined by $\theta$, each one with $0.1$ and $0.9$ of the total mass, and similarly $\theta_{\#}\mu^2$ is concentrated at two points (blue triangles) each one with $0.5$ of the total mass.*

Now, let us prove Lemma 2.3, that is, let us show that each measure $\gamma_\theta^{\mu^1,\mu^2}$ defined in (9) is a transport plan in $\Gamma(\mu^1, \mu^2)$.

*Proof of Lemma 2.3.* Let $x \in \mathbb{R}^d$. First, if $p(x) = 0$, then $u_\theta^{\mu^1,\mu^2}(x,y) = 0$ for every $y \in \mathbb{R}^d$, and so $\gamma_\theta^{\mu^1,\mu^2}(\{x\} \times \mathbb{R}^d) = 0 = p(x) = \mu^1(\{x\})$. Now, assume that $p(x) \neq 0$, then

$$\sum_{y \in \mathbb{R}^d} u_\theta^{\mu^1,\mu^2}(x,y) = \frac{p(x)}{P(\bar{x}^\theta)} \sum_{y \in \mathbb{R}^d : q(y) \neq 0} \frac{q(y)}{Q(\bar{y}^\theta)} \Lambda_\theta^{\mu^1,\mu^2}(\{(\bar{x}^\theta, \bar{y}^\theta)\})$$

$$= \frac{p(x)}{P(\bar{x}^\theta)} \sum_{\bar{y}^\theta \in \mathbb{R}^d/\sim_\theta : Q(\bar{y}^\theta) \neq 0} \left(\sum_{y \in \bar{y}^\theta} q(y)\right) \frac{1}{Q(\bar{y}^\theta)} \Lambda_\theta^{\mu^1,\mu^2}(\{(\bar{x}^\theta, \bar{y}^\theta)\})$$

$$= \frac{p(x)}{P(\bar{x}^\theta)} \sum_{\bar{y}^\theta \in \mathbb{R}^d/\sim_\theta : Q(\bar{y}^\theta) \neq 0} Q(\bar{y}^\theta) \frac{1}{Q(\bar{y}^\theta)} \Lambda_\theta^{\mu^1,\mu^2}(\{(\bar{x}^\theta, \bar{y}^\theta)\})$$

$$= \frac{p(x)}{P(\bar{x}^\theta)} \sum_{\bar{y}^\theta \in \mathbb{R}^d/\sim_\theta} \Lambda_\theta^{\mu^1,\mu^2}(\{(\bar{x}^\theta, \bar{y}^\theta)\}) = \frac{p(x)}{P(\bar{x}^\theta)} P(\bar{x}^\theta) = p(x).$$

Thus, $\gamma_\theta^{\mu^1,\mu^2}(\{x\} \times \mathbb{R}^d) = p(x) = \mu^1(\{x\})$ for every $x \in \mathbb{R}^d$. Similarly, $\sum_{x \in \mathbb{R}^d} u_\theta^{\mu^1,\mu^2}(x,y) = q(y)$, or equivalently, $\gamma_\theta^{\mu^1,\mu^2}(\mathbb{R}^d \times \{y\}) = q(x) = \mu^2(\{y\})$ for every $y \in \mathbb{R}^d$.  □

**Remark A.2** (Expected Sliced Transport for uniform discrete measures). *Let $\mu^1, \mu^2 \in \mathcal{P}_{(N)}(\mathbb{R}^d)$ of the form $\mu^1 = \frac{1}{N} \sum_{i=1}^N \delta_{x_i}, \mu^2 = \frac{1}{N} \sum_{j=1}^N \delta_{y_j}$. Then, the expected sliced transport plan between $\mu^1$ and $\mu^2$, $\bar{\gamma}^{\mu^1,\mu^2} = \mathbb{E}_{\theta \sim \sigma}[\gamma_\theta^{\mu^1,\mu^2}]$, defines a discrete measure on $\mathcal{P}(\mathbb{R}^d \times \mathbb{R}^d)$ supported on $\{(x_i, y_j)\}_{i,j \in [N]}$ where it takes the values*

$$\bar{\gamma}^{\mu^1,\mu^2}(\{x_i, y_j\}) = \int_{\mathbb{S}^{d-1}} \gamma_\theta^{\mu^1,\mu^2}(\{x_i, y_j\}) d\sigma(\theta) \qquad \forall i \in [N], j \in [N]. \tag{18}$$

*Thus, it can be regarded as an $N \times N$ matrix whose $(i, j)$-entry is given by (18). Moreover, each $N \times N$ matrix $u_\theta^{\mu^1,\mu^2}$ defined by (6) can be obtained by swapping rows from the $N \times N$ identity matrix multiplied by $1/N$, there are finitely many matrices (precisely, $N!$ matrices in total). Hence, the function $\theta \mapsto u_\theta^{\mu^1,\mu^2}$ is a piece-wise constant matrix-valued function. Thus, the function $\theta \mapsto \gamma_\theta^{\mu^1,\mu^2}$*

*(where $\gamma_\theta^{\mu^1,\mu^2}$ is an in (8)) is a measurable function. This can be generalized for any pair of finite discrete measures as in the following remarks.*

**Remark A.3** (Expected Sliced Transport for finite discrete measures). *Consider arbitrary finite discrete measures $\mu^1 = \sum_{i=1}^n p(x_i)\delta_{x_i}$ and $\mu^2 = \sum_{j=1}^m q(y_i)\delta_{y_j}$, i.e., discrete measures with finite support.*

- *Fix $x_i, y_j \in \mathbb{R}^d$, then $\theta \mapsto \frac{p(x_i)q(y_j)}{P(\bar{x}_i^\theta)Q(\bar{y}_j^\theta)} \neq 1$ for all but a finite number of directions. This is due to the fact that only for finitely many directions $\theta$ we obtain overlaps of the projected points $\theta \cdot x$.*

- *The optimal transport plan $\Lambda_\theta^{\mu^1,\mu^2}$ is given by "matching from left to right until fulfilling the target bins": that is, one has to order the points similarly as in (3) and consider an "increasing" assignment plan. Since the order of $\{\theta \cdot x_i\}_{i=1}^n$ and $\{\theta \cdot y_j\}_{j=1}^m$ changes a finite number of times when varying $\theta \in \mathbb{S}^{d-1}$, the function $\theta \mapsto \Lambda_\theta^{\mu^1,\mu^2}$ takes a finite number of possible transport plan options.*

*Thus, the range of $\theta \mapsto u_\theta^{\mu^1,\mu^2}$ is finite.*

**Remark A.4** ($\bar{\gamma}^{\mu^1,\mu^2}$ is well-defined for finite discrete measures). *First, we notice that for each $\theta$, the support of $\Lambda_\theta^{\mu^1,\mu^2}$ is finite or countable, and so the support of $\gamma_\theta^{\mu^1,\mu^2}$ is also finite or countable. Given an arbitrary point $(x,y) \in \mathbb{R}^d \times \mathbb{R}^d$, we have to justify that the function $\mathbb{S}^{d-1} \ni \theta \mapsto u_\theta^{\mu^1,\mu^2}(x,y)$ is (Borel)-measurable: If the supports of $\mu^1$ and $\mu^2$ are finite, by Remark A.3, $\theta \mapsto u_\theta^{\mu^1,\mu^2}$ is a piece-wise constant function, and so it is measurable and integrable on the sphere. For the general case, when the supports of $\mu^1$ and $\mu^2$ are countable, we refer to Remark A.14.*

**Lemma A.5** ($\bar{\gamma}^{\mu^1,\mu^2}$ is a transport plan between $\mu^1$ and $\mu^2$). *We have that $\bar{\gamma}^{\mu^1,\mu^2} \in \Gamma(\mu^1,\mu^2)$, i.e., it has marginals $\mu^1$ and $\mu^2$. This is because for each $\theta \in \mathbb{S}^{d-1}$, $\gamma_\theta^{\mu^1,\mu^2} \in \Gamma(\mu^1,\mu^2)$ and because $\sigma$ is a probability measure on $\mathbb{S}^{d-1}$. Then, $\bar{\gamma}^{\mu^1,\mu^2}$ is a convex combination of transport plans $\gamma_\theta^{\mu^1,\mu^2}$, and since $\Gamma(\mu^1,\mu^2)$ is a convex set, we obtain that $\bar{\gamma}^{\mu^1,\mu^2} \in \Gamma(\mu^1,\mu^2)$. Precisely, for every test function $\phi : \mathbb{R}^d \to \mathbb{R}$*

$$\int_{\mathbb{R}^d \times \mathbb{R}^d} \phi(x)d\bar{\gamma}^{\mu^1,\mu^2}(x,y) = \int_{\mathbb{S}^{d-1}} \int_{\mathbb{R}^d \times \mathbb{R}^d} \phi(x)d\gamma_\theta^{\mu^1,\mu^2}(x,y)d\sigma(\theta)$$

$$= \int_{\mathbb{S}^{d-1}} \int_{\mathbb{R}^d} \phi(x)d\mu^1(x)d\sigma(\theta) = \int_{\mathbb{R}^d} \phi(x)d\mu^1(x)$$

*Similarly, $\int_{\mathbb{R}^d \times \mathbb{R}^d} \psi(y)d\bar{\gamma}^{\mu^1,\mu^2}(x,y) = \int_{\mathbb{R}^d} \psi(y)d\mu^2(y)$, and so $\bar{\gamma}^{\mu^1,\mu^2}$ has marginals $\mu^1$ and $\mu^2$.*

## A.2 AN AUXILIARY RESULT

For simplicity, in this paper we consider the strictly convex cost $\|x-y\|^p$ ($1 < p < \infty$). Also, in this section we consider $\sigma = \mathcal{U}(\mathbb{S}^{d-1})$ and in this case we denote $d\sigma(\theta) = d\theta$.

**Proposition A.6.** *Let $\Omega \subset \mathbb{R}^d$ be a compact set, and let $\mu^1, \mu^2 \in \mathcal{P}(\Omega)$. Let $(\mu_n^1)_{n\in\mathbb{N}}, (\mu_n^2)_{n\in\mathbb{N}} \subset \mathcal{P}(\Omega)$ be sequences such that, for $i = 1, 2$, $\mu_n^i \rightharpoonup^* \mu^i$ as $n \to \infty$, where the limit is in the weak*-topology. For each $n \in \mathbb{N}$, consider optimal transport plans $\gamma_n \in \Gamma^*(\mu_n^1, \mu_n^2)$. Then, there exists a subsequence such that $\gamma_{n_k} \rightharpoonup^* \gamma$, for some optimal transport plan $\gamma \in \Gamma^*(\mu^1, \mu^2)$.*

*Proof.* As $(\gamma_n)_{n\in\mathbb{N}}$ is a sequence of probability measures, their mass is 1, by Banach-Alaoglu Theorem, there exists a subsequence such that $\gamma_{n_k} \rightharpoonup^* \gamma$, for some $\gamma \in \mathcal{P}(\Omega \times \Omega)$. It is easy to see that the limit $\gamma$ has marginals $\mu^1, \mu^2$. Indeed, given any test functions $\phi, \psi \in C(\Omega)$, since each $\gamma_n$ has marginals $\mu_n^1, \mu_n^2$, we have

$$\int_{\Omega \times \Omega} \phi(x)d\gamma_n(x,y) = \int_\Omega \phi(x)d\mu_n^1(x) \quad \text{and} \quad \int_{\Omega \times \Omega} \psi(y)d\gamma_n(x,y) = \int_\Omega \psi(y)d\mu_n^2(y)$$

and taking limit as $n \to \infty$, we obtain

$$\int_{\Omega \times \Omega} \phi(x) d\gamma(x, y) = \int_{\Omega} \phi(x) d\mu^1(x) \quad \text{and} \quad \int_{\Omega \times \Omega} \psi(y) d\gamma(x, y) = \int_{\Omega} \psi(y) d\mu^2(y).$$

Now, we only have to prove the optimality of $\gamma$ for the OT problem between $\mu^1$ and $\mu^2$. Since $(x, y) \mapsto \|x - y\|^p$ is continuous and $\gamma_n$, $\gamma$ are compactly supported, by using that for each $n \in \mathbb{N}$, $\gamma_n$ is optimal for the OT problem between $\mu_n^1$ and $\mu_n^2$, we have

$$\lim_{n \to \infty} \left( W_p(\mu_n^1, \mu_n^2) \right)^p = \lim_{n \to \infty} \int_{\Omega \times \Omega} \|x - y\|^p d\gamma_n(x, y)$$

$$= \int_{\Omega \times \Omega} \|x - y\|^p d\gamma(x, y) \geq \left( W_p(\mu^1, \mu^2) \right)^p \quad (19)$$

Also, by hypothesis and (Santambrogio, 2015, Theorem 5.10) we have that for any $\nu \in \mathcal{P}(\Omega)$,

$$\lim_{n \to \infty} W_p(\mu_n^1, \nu) = W_p(\mu^1, \nu) \quad \text{and} \quad \lim_{n \to \infty} W_p(\nu, \mu_n^2) = W_p(\nu, \mu^2).$$

So, by using the the triangle inequality for the $p$-Wasserstein distance we get

$$\lim_{n \to \infty} W_p(\mu_n^1, \mu_n^2) \leq \lim_{n \to \infty} W_p(\mu_n^1, \mu^1) + \lim_{n \to \infty} W_p(\mu^2, \mu_n^2) + W_p(\mu^1, \mu^2)$$

$$= 0 + W_p(\mu^1, \mu^2) = W_p(\mu^1, \mu^2) \quad (20)$$

Therefore, from (20) and (19) we have that

$$\lim_{n \to \infty} W_p(\mu_n^1, \mu_n^2) = W_p(\mu^1, \mu^2).$$

In particular, in (19) we have that the following equality holds:

$$\int_{\Omega \times \Omega} \|x - y\|^p d\gamma(x, y) = \left( W_p(\mu^1, \mu^2) \right)^p.$$

As a result, $\gamma$ is optimal for the OT problem between $\mu^1$ and $\mu^2$. $\qquad \square$

### A.3 Finite discrete measures with rational weights

Let us denote by $\mathcal{P}_{\mathbb{Q}}(\mathbb{R}^d)$ the set of finite discrete probability measures in $\mathbb{R}^d$ with rational weights, that is, $\mu \in \mathcal{P}_{\mathbb{Q}}(\mathbb{R}^d)$ if and only if it is of the form $\mu = \sum_{i=1}^m q_i \delta_{x_i}$ with $x_i \in \mathbb{R}^d$, $q_i \in \mathbb{Q}$ $\forall i \in [m]$ for some $m \in \mathbb{N}$, and $\sum_{i=1}^m q_i = 1$. We have

$$\mathcal{P}_{(N)}(\mathbb{R}^d) \subset \mathcal{P}_{\mathbb{Q}}(\mathbb{R}^d), \qquad \forall N \in \mathbb{N}.$$

In the definition of an uniform discrete measure $\mu = \frac{1}{N} \sum_{i=1}^N \delta_{x_i} \in \mathcal{P}_{(N)}(\mathbb{R}^d)$ one can allow $x_i = x_j$ for some pairs of indexes $i \neq j$.

**Proposition A.7.** $\mathcal{D}_p(\cdot, \cdot)$ defined by (11) is a metric in $\mathcal{P}_{\mathbb{Q}}(\mathbb{R}^d)$.

*Proof.* This was essentially pointed out Remark 2.9: When restricting to the space $\mathcal{P}_{(N)}(\mathbb{R}^d)$, our $\mathcal{D}_p(\cdot, \cdot)$ and the Projected Wasserstein distance presented in (Rowland et al., 2019) coincide. Rowland et al. (2019) prove the metric property. We recall here their main argument, which is used for showing the triangle inequality. Given $\mu^1, \mu^2, \mu^3 \in \mathcal{P}_{(N)}(\mathbb{R}^d)$ of the form $\mu^1 = \frac{1}{N} \sum_{i=1}^N \delta_{x_i}$, $\mu^2 = \frac{1}{N} \sum_{i=1}^N \delta_{y_i}$, $\mu^3 = \frac{1}{N} \sum_{i=1}^N \delta_{z_i}$. Fix $\theta \in \mathbb{S}^{d-1}$, and consider permutations $\zeta_\theta, \tau_\theta, \xi_\theta \in \mathbf{S}_N$, so that

$$\theta \cdot x_{\zeta_\theta^{-1}(1)} \leq \cdots \leq \theta \cdot x_{\zeta_\theta^{-1}(N)},$$

$$\theta \cdot y_{\tau_\theta^{-1}(1)} \leq \cdots \leq \theta \cdot y_{\tau_\theta^{-1}(N)},$$

$$\theta \cdot z_{\xi_\theta^{-1}(1)} \leq \cdots \leq \theta \cdot z_{\xi_\theta^{-1}(N)}$$

Thus, the key idea is that

$$\mathcal{D}_p(\mu^1, \mu^2)^p = \int_{\mathbb{S}^{d-1}} \frac{1}{N} \sum_{i=1}^{N} \|x_{\zeta_\theta^{-1}(i)} - y_{\tau_\theta^{-1}(i)}\|^p d\theta = \sum_{\zeta,\tau,\xi \in \mathbf{S}_N} \frac{\mathbf{q}(\zeta,\tau,\xi)}{N} \sum_{i=1}^{N} \|x_{\zeta^{-1}(i)} - y_{\tau^{-1}(i)}\|^p$$

$$\mathcal{D}_p(\mu^2, \mu^3)^p = \int_{\mathbb{S}^{d-1}} \frac{1}{N} \sum_{i=1}^{N} \|y_{\tau_\theta^{-1}(i)} - z_{\xi_\theta^{-1}(i)}\|^p d\theta = \sum_{\zeta,\tau,\xi \in \mathbf{S}_N} \frac{\mathbf{q}(\zeta,\tau,\xi)}{N} \sum_{i=1}^{N} \|y_{\tau^{-1}(i)} - z_{\xi^{-1}(i)}\|^p$$

$$\mathcal{D}_p(\mu^3, \mu^1)^p = \int_{\mathbb{S}^{d-1}} \frac{1}{N} \sum_{i=1}^{N} \|z_{\xi_\theta^{-1}(i)} - x_{\zeta_\theta^{-1}(i)}\|^p d\theta = \sum_{\zeta,\tau,\xi \in \mathbf{S}_N} \frac{\mathbf{q}(\zeta,\tau,\xi)}{N} \sum_{i=1}^{N} \|z_{\xi^{-1}(i)} - x_{\zeta^{-1}(i)}\|^p$$

where $\mathbf{q} \in \mathcal{P}(\mathbf{S}_N \times \mathbf{S}_N \times \mathbf{S}_N)$ is such that $\mathbf{q}(\zeta,\tau,\xi)$ is the probability that the tuple permutations $(\zeta,\tau,\xi) = (\zeta_\theta, \tau_\theta, \xi_\theta)$ are required, given that $\theta$ is drawn from $\mathrm{Unif}(\mathbb{S}^{d-1})$. With these alternative expressions established by the authors in (Rowland et al., 2019), the triangle inequality follows from the standard Minkowski inequality for weighted finite $L^p$-spaces.

Finally, notice that we have used the fact that each $\mu^1$ is associated to $N$-indexes $\{1, \ldots, N\}$, without asking that the points $\{x_i\}$ do not overlap, i.e., they could be repeated. That is, given $\mu^1 = \frac{1}{N}\sum_{i=1}^{n} \delta_{x_i} \in \mathcal{P}_{(N)}(\mathbb{R}^d)$ one can allow $x_i = x_j$ for some pairs of indexes $i \neq j$ (analogously for $\mu^2$ and $\mu^3$). Thus, the proof also holds for measures in $\mathcal{P}_{\mathbb{Q}}(\mathbb{R}^d)$: Indeed, let $\mu^1, \mu^2, \mu^3 \in \mathcal{P}_{\mathbb{Q}}(\mathbb{R}^d)$ be of the form $\mu^1 = \sum_{i=1}^{n_1} \frac{r_i^1}{s_i^1}\delta_{x_i}$, $\mu^2 = \sum_{i=1}^{n_2} \frac{r_i^2}{s_i^2}\delta_{y_i}$, $\mu^3 = \sum_{i=1}^{n_3} \frac{r_i^3}{s_i^3}\delta_{z_i}$ with $r_i^j, q_i^j \in \mathbb{N}$. First, consider the $n_1'$ as the least common multiple of $\{s_1^1, \ldots, s_{n_1}^1\}$ and, for each $i \in [n_1]$, let $\tilde{r}_i^1$ so that $\frac{\tilde{r}_i^1}{n_1'} = \frac{r_i^1}{s_i^1}$. Thus, we can rewrite $\mu^1 = \sum_{i=1}^{n_1} \frac{\tilde{r}_i^1}{n_1'}\delta_{x_i}$. Notice that, since $\mu^1$ is a probability measure, we have $n_1' = \sum_{i=1}^{n_1} \tilde{r}_i^1$. Now, for each $i \in [n_1]$ such that $\tilde{r}_i^1 > 1$, consider $\tilde{r}_i^1$ copies of the corresponding point $x_i$ so that we can rewrite $\mu^1 = \sum_{i=1}^{n_1'} \frac{1}{n_1'}\delta_{x_i}$ (where we recall that $n_1' = \sum_{i=1}^{n_1} \tilde{r}_i^1$ and the points $x_i$ in the new expression can be repeated, i.e., they are not necessarily all different). Repeat this process to rewrite $\mu^2 = \sum_{i=1}^{n_2'} \frac{1}{n_2'}\delta_{y_i}$, $\mu^3 = \sum_{i=1}^{n_3'} \frac{1}{n_3'}\delta_{z_i}$. Now, consider $N$ as the least common multiple of $n_1', n_2', n_3'$, and rewrite the measures as $\mu^1 = \frac{1}{N}\sum_{i=1}^{N} \delta_{x_i}$, $\mu^2 = \frac{1}{N}\sum_{i=1}^{N} \delta_{y_i}$, $\mu^3 = \frac{1}{N}\sum_{i=1}^{N} \delta_{z_i}$ where the points $x_i, y_i, z_i$ can be repeated if needed. Thus, $\mu^1, \mu^2, \mu^3$ can be regarded as measures in $\mathcal{P}_{(N)}(\mathbb{R}^d)$ where $\mathcal{D}_p(\cdot, \cdot)$ behaves as a metric. $\square$

## A.4 THE PROOF FOR GENERAL FINITE DISCRETE MEASURES

We first introduce some notation. Consider a finite discrete probability measure $\mu \in \mathcal{P}(\mathbb{R}^d)$ of the form $\mu = \sum_{i=1}^{m} p^i \delta_{x_i}$, with general weights $p^i \in \mathbb{R}_+$ such that $\sum_{i=1}^{m} p^i = 1$. For each $i \in \{1, \ldots, m-1\}$, consider an increasing sequence of rational numbers $(p_n^i)_{n \in \mathbb{N}} \subset \mathbb{Q}$, with $0 \leq p_n^i \leq p^i$, such that $\lim_{n \to \infty} p_n^i = p^i$. For $i = m$, consider the sequence $(p_n^m)_{n \in \mathbb{N}} \subset \mathbb{Q}$ defined by $0 \leq p_n^m := 1 - \sum_{i=1}^{m-1} p_n^i \leq 1$. Thus, $\lim_{n \to \infty} p_n^m = 1 - \lim_{n \to \infty} \sum_{i=1}^{m-1} p_n^i = 1 - \sum_{i=1}^{m-1} p^i = p^m$. Define the sequence of probability measures $(\mu_n)_{n \in \mathbb{N}}$ given by $\mu_n := \sum_{i=1}^{m} p_n^i \delta_{x_i} \in \mathcal{P}_{\mathbb{Q}}(\mathbb{R}^d)$. It is easy to show that $(\mu_n)_{n \in \mathbb{N}}$ converges to $\mu$ in Total Variation (i.e., uniform convergence or strong convergence): Indeed, let $\varepsilon > 0$. For each $i \in [m]$, let $N_i \in \mathbb{N}$ such that $|p_n^i - p^i| < \varepsilon/m \; \forall n \geq N_i$ and define $N = \max\{N_1, \ldots, N_m\}$. Now, given any set $B \subset \mathbb{R}^d$ we obtain, for $n \geq N$,

$$
\begin{aligned}
|\mu_n(B) - \mu(B)| &= \left| \sum_{i \in [m]: x_i \in B} (p_n^i - p^i) \right| \quad &&(\mu_n \text{ and } \mu \text{ have the same support}) \\
&\leq \sum_{i \in [m]: x_i \in B} |p_n^i - p^i| \quad &&(\text{triangle inequality}) \\
&\leq \sum_{i \in [m]} |p_n^i - p^i| \quad &&(\text{sum over all indexes to get independence of the set } B) \\
&< \varepsilon.
\end{aligned}
$$

This shows that $\lim_{n \to \infty} \|\mu_n - \mu\|_{\mathrm{TV}} = 0$. Moreover, this shows that in this case, i.e., when approximating a finite discrete measure $\mu$ by a sequence of measures having the same support as $\mu$, we only care about point-wise convergence.

We will now introduce some lemmas which together with the above proposition will allow us to prove the metric property of $\mathcal{D}_p(\cdot, \cdot)$ for finite discrete probability measures. For all of them we will consider:

- $\mu^1, \mu^2$ two finite discrete probability measures in $\mathbb{R}^d$ given by $\mu^1 = \sum_{i=1}^{m_1} p^i \delta_{x_i}$, $\mu^2 = \sum_{j=1}^{m_2} q^j \delta_{y_j}$

- $(\mu_n^1)_{n \in \mathbb{N}}$, $(\mu_n^2)_{n \in \mathbb{N}}$ approximating sequences of probability measures $\mu_n^1 = \sum_{i=1}^{m_1} p_n^i \delta_{x_i}$, $\mu_n^2 = \sum_{j=1}^{m_2} q_n^j \delta_{y_j}$, with rational weights $\{p_n^i\}, \{q_n^j\}$, defined in analogy to what we have already done, i.e., so that, for $k = 1, 2$ we have that $(\mu_n^k)_{n \in \mathbb{N}}$ is a sequence of probability measures that converges to $\mu^k$ in Total Variation).

Also, for each $\theta$, $\Lambda_\theta^{\mu^1, \mu^2}$ denotes the unique optimal transport plan between $\theta_\# \mu^1$ and $\theta_\# \mu^2$; $\gamma_\theta^{\mu^1, \mu^2}$ denotes lifted transport plan between $\mu^1$ and $\mu^2$ given as in (9); and $\bar{\gamma}^{\mu^1, \mu^2}$ the expected sliced transport plan between $\mu^1$ and $\mu^2$ given as in (10). Similarly, for each $n \in \mathbb{N}$ we consider the plans $\Lambda_\theta^{\mu_n^1, \mu_n^2}$, $\gamma_\theta^{\mu_n^1, \mu_n^2}$, and $\bar{\gamma}^{\mu_n^1, \mu_n^2}$.

**Lemma A.8.** *The sequence* $\left(\Lambda_\theta^{\mu_n^1, \mu_n^2}\right)_{n \in \mathbb{N}}$ *converges to* $\Lambda_\theta^{\mu^1, \mu^2}$ *in Total Variation.*

**Lemma A.9.** *The sequence* $\left(\gamma_\theta^{\mu_n^1, \mu_n^2}\right)_{n \in \mathbb{N}}$ *converges to* $\gamma_\theta^{\mu^1, \mu^2}$ *in Total Variation.*

**Lemma A.10.** *The sequence* $\left(\bar{\gamma}^{\mu_n^1, \mu_n^2}\right)_{n \in \mathbb{N}}$ *converges to* $\bar{\gamma}^{\mu^1, \mu^2}$ *in Total Variation.*

**Lemma A.11.** $\lim_{n \to \infty} \mathcal{D}_p(\mu_n^1, \mu_n^2) = \mathcal{D}_p(\mu^1, \mu^2)$.

In general, notice that since for every $i \in [m_1]$, $j \in [m_2]$ we have that $\lim_{n \to \infty} p_n^i = p^i$, and $\lim_{n \to \infty} q_n^j = p^j$, then we obtain that $\lim_{n \to \infty} P_n^i = P^i$, and $\lim_{n \to \infty} Q_n^j = Q^j$, where $P^i = \sum_{i \in [m_1]: x_i \in \bar{x}_i^\theta} p^i$, $Q^j = \sum_{j \in [m_2]: y_j \in \bar{y}_j^\theta} q^j$, and where, for each $n \in \mathbb{N}$, $P_n^i$ and $Q_n^j$ are analogously defined (see Subsection 2.2.2). Thus,

$$\lim_{n \to \infty} \frac{p_n^i q_n^j}{P_n^i Q_n^j} = \frac{p^i q^j}{P^i Q^j} \qquad \forall i \in [m_1], j \in [m_2]. \tag{21}$$

*Proof of Lemma A.8.* The support of all the measures we are considering are finite and so, the measures have compact support. Hence, we can apply Proposition A.6 to $\theta_\# \mu^i$, $(\theta_\# \mu^i)_{n \in \mathbb{N}}$, $i = 1, 2$. Specifically, given $(\Lambda_\theta^{\mu_n^1, \mu_n^2})_{n \in \mathbb{N}} \in \Gamma^*(\theta_\# \mu_n^1, \theta_\# \mu_n^2)$, there exists a subsequence $(\Lambda_\theta^{\mu_{n_k}^1, \mu_{n_k}^2})_{K \in \mathbb{N}}$ and $\Lambda_\theta \in \Gamma^*(\theta_\# \mu^1, \theta_\# \mu^2)$ such that

$$\Lambda_\theta^{\mu_{n_k}^1, \mu_{n_k}^2} \rightharpoonup^* \Lambda_\theta. \tag{22}$$

As we are in one dimension, the set $\Gamma^*(\theta_\# \mu^1, \theta_\# \mu^2)$ is a singleton, and so we have that $\Lambda_\theta = \Lambda_\theta^{\mu^1, \mu^2}$ is the unique optimal transport plan. Since the supports of all the measures are the same, (that is, $\{(\theta \cdot x_i, \theta \cdot y_j)\}_{i \in [m_1], j \in [m_2]}$), the weak$^*$ convergence in (22) implies the stronger convergence in Total Variation.

Now, suppose that the original sequence $(\Lambda_\theta^{\mu_n^1, \mu_n^2})_{n \in \mathbb{N}}$ does not converge to $\Lambda_\theta^{\mu^1, \mu^2}$ (in Total Variation). Then, given $\varepsilon > 0$, there exists a subsequence $(\Lambda_\theta^{\mu_{n_j}^1, \mu_{n_j}^2})_{j \in \mathbb{N}}$ such that

$$\|\Lambda_\theta^{\mu_{n_j}^1, \mu_{n_j}^2} - \Lambda_\theta^{\mu^1, \mu^2}\|_{\text{TV}} > \varepsilon \tag{23}$$

But again, from Proposition A.6, using that the supports of all the measures involved are the same set $\{(\theta \cdot x_i, \theta \cdot y_j)\}_{i \in [m_1], j \in [m_2]}$, and the fact that $\Gamma^*(\theta_\# \mu^1, \theta_\# \mu^2) = \{\Lambda_\theta^{\mu^1, \mu^2}\}$ (only one optimal transport plan), we have that there exists a sub-subsequence such that

$$\|\Lambda_\theta^{\mu_{n_{j_i}}^1, \mu_{n_{j_i}}^2} - \Lambda_\theta^{\mu^1, \mu^2}\|_{\text{TV}} < \varepsilon.$$

contradicting (23). Since the contradiction is achieved from assuming that the whole sequence $(\Lambda_\theta^{\mu_n^1, \mu_n^2})_{n \in \mathbb{N}}$ does not converge to $\Lambda_\theta^{\mu^1, \mu^2}$, we have that, in fact, it does converge to $\Lambda_\theta^{\mu^1, \mu^2}$ in Total Variation. $\qquad \square$

*Proof of Lemma A.9.* This holds by looking at (9): Due to the fact that the supports of $\mu_n^1$ and $\mu^1$ are the same (respectively, for $\mu_n^2$ and $\mu^2$), we only care about the locations $\{(x_i, y_j)\}_{i \in [m_1], j \in [m_2]}$, and then by using (21) and the convergence from Lemma A.8, the result holds true. $\square$

*Proof of Lemma A.10.* As pointed out before, we only care about point-wise convergence: That is, since the supports of the measures involved coincide (are the same set $\{(x_i, y_j)\}_{i \in [m_1], j \in [m_2]}$) weak* convergence, point-wise convergence and convergence in Total Variation are equivalent.

Since $0 \leq \gamma_\theta^{\mu_n^1, \mu_n^2}(\{(x_i, y_j)\}) \leq 1$ and $\mathbb{S}^{d-1}$ is compact, by the convergence result from Lemma A.9 and using the Dominated Convergence Theorem, we have that for each $i \in [m_1], j \in [m_2]$,

$$
\begin{aligned}
\lim_{n \to \infty} \bar{\gamma}^{\mu_n^1, \mu_n^2}(\{(x_i, y_j)\}) &= \lim_{n \to \infty} \int_{\mathbb{S}^{d-1}} \gamma_\theta^{\mu_n^1, \mu_n^2}(\{(x_i, y_j)\}) d\theta \\
&= \int_{\mathbb{S}^{d-1}} \lim_{n \to \infty} \gamma_\theta^{\mu_n^1, \mu_n^2}(\{(x_i, y_j)\}) d\theta \\
&= \int_{\mathbb{S}^{d-1}} \gamma_\theta^{\mu^1, \mu^2}(\{(x_i, y_j)\}) d\theta = \bar{\gamma}^{\mu^1, \mu^2}(\{(x_i, y_j)\}) \qquad \square
\end{aligned}
$$

*Proof of Lemma A.11.*

$$
\left| \mathcal{D}_p(\mu_n^1, \mu_n^2)^p - \mathcal{D}_p(\mu^1, \mu^2)^p \right| \leq \max_{i \in [m_1], j \in [m_2]} \{\|x_i - y_j\|^p\} \|\bar{\gamma}^{\mu_n^1, \mu_n^2} - \bar{\gamma}^{\mu^1, \mu^2}\|_{\text{TV}} \quad (24)
$$

where the RHS goes to 0 as $n \to \infty$, due to Lemma A.10. $\square$

**Theorem A.12.** *$\mathcal{D}_p(\cdot, \cdot)$ is a metric for the space of finite discrete probability measures in $\mathbb{R}^d$.*

*Proof.*

- Symmetry: The way we constructed $\mathcal{D}_p(\cdot, \cdot)$ makes it so that $\mathcal{D}_p(\mu^1, \mu^2) = \mathcal{D}_p(\mu^2, \mu^1)$.

- Positivity: It is clear that by definition $\mathcal{D}_p(\mu^1, \mu^2) \geq 0$.

- Identity of indiscernibles:

  First, if $\mu^1 = \mu^2 =: \mu$, then $\gamma_\theta^{\mu^1, \mu^2} = (id \times id)_\# \mu$ for all $\theta \in \mathbb{S}^{d-1}$. Hence $\bar{\gamma}^{\mu^1, \mu^2} = (id \times id)_\# \mu$ which implies $\mathcal{D}_p(\mu^1, \mu^2) = 0$.

  Secondly, if $\mu^1, \mu^2$ are such that $\mathcal{D}_p(\mu^1, \mu^2) = 0$, by having

  $$
  W_p(\mu^1, \mu^2) \leq \mathcal{D}_p(\mu^1, \mu^2) = 0,
  $$

  we can use the fact that $W_p(\cdot, \cdot)$ satisfies the identity of indiscernibles by being a distance. That is, $W_p(\mu^1, \mu^2) = 0$ implies $\mu^1 = \mu^2$.

- Triangle inequality: Given $\mu^1, \mu^2, \mu^3$ arbitrary finite discrete measures with arbitrary real weights, consider approximating sequences $(\mu_n^1)_{n \in \mathbb{N}}, (\mu_n^2)_{n \in \mathbb{N}}, (\mu_n^3)_{n \in \mathbb{N}}$ in $\mathcal{P}_{\mathbb{Q}}(\mathbb{R}^d)$ as before. Notice that every subsequence of $(\mu_n^1)_{n \in \mathbb{N}}$ (respectively of $(\mu_n^2)_{n \in \mathbb{N}}$ and $(\mu_n^3)_{n \in \mathbb{N}}$) will converge to $\mu^1$ (respectively, to $\mu^2$ and $\mu^3$), as every subsequence of a convergent sequence is convergent.

  By Proposition A.7, we have that, for each $n \in \mathbb{R}^d$,

  $$
  \mathcal{D}_p(\mu_n^1, \mu_n^2) \leq \mathcal{D}_p(\mu_n^1, \mu_n^3) + \mathcal{D}_p(\mu_n^3, \mu_n^2).
  $$

  Taking the limit as $n \to \infty$, from Lemma (A.11) we obtain

  $$
  \mathcal{D}_p(\mu^1, \mu^2) \leq \mathcal{D}_p(\mu^1, \mu^3) + \mathcal{D}_p(\mu^3, \mu^2). \qquad \square
  $$

## A.5 DISCRETE MEASURES WITH COUNTABLE SUPPORT

**Lemma A.13.** *Let* $\mu = \sum_{m \in \mathbb{N}}^{\infty} p^m \delta_{x_m}$ *be a discrete probability measure with countable support* $\{x_m\}_{m \in \mathbb{N}}$. *Let* $\sigma$ *be an absolutely continuous probability measure with respect to the Lebesgue measure on the sphere (we write,* $\sigma \ll \mathrm{Unif}(\mathbb{S}^{d-1})$). *Let*

$$S_\mu := \{\theta \in \mathbb{S}^{d-1} : \theta \cdot x_m = \theta \cdot x_{m'} \text{ for some pair } (m, m') \text{ with } m \neq m'\}.$$

*Then*

$$\sigma(S_\mu) = 0.$$

*Proof.* First, consider distinct points $x_m, x_{m'}$ on the support of $\mu$, and let

$$S(x_m, x_{m'}) = \{\theta \in \mathbb{S}^{d-1} : \theta \cdot x_m = \theta \cdot x_{m'}\}.$$

It is straightforward to verify that

$$S(x_m, x_{m'}) = \mathbb{S}^{d-1} \cap \mathrm{span}(\{x_m - x_{m'}\})^{\perp},$$

where $\mathrm{span}(\{x_m - x_{m'}\})^{\perp}$ is the orthogonal subspace to the line in the direction of the vector $(x_m - x_{m'})$. Thus, $S(x_m, x_{m'})$ is a subset of a $d-2$-dimensional sub-sphere in $\mathbb{S}^{d-1}$, and therefore

$$\sigma_{\mathbb{S}^{d-1}}(S(x_m, x_{m'})) = 0 \tag{25}$$

Since,

$$S_\mu = \bigcup_{(x_m, x_{m'}) \in \mathbf{M}} S(x_m, x_{m'}), \qquad \text{where} \quad \mathbf{M} = \{(x_m, x_{m'}) : m \neq m'\} \tag{26}$$

we have that

$$\sigma(S_\mu) \leq \sum_{(x_i, x_i') \in \mathbf{M}} \sigma(S(x_m, x_{m'})) = 0.$$

since $\mathbf{M}$ is countable (indeed, $|\mathbf{M}| \leq |\mathrm{supp}(\mu) \times \mathrm{supp}(\mu)|$). $\qquad \square$

**Remark A.14.** *[$\bar{\gamma}^{\mu^1, \mu^2}$ is well-defined for discrete measures with countable support] Let* $\sigma \ll \mathrm{Unif}(\mathbb{S}^{d-1})$. *Given two discrete probability measures* $\mu^1 = \sum p(x)\delta_x$ *and* $\mu^2 = \sum q(y)\delta_y$ *with countable support, from Lemma A.13, we have that* $\sigma(S_{\mu^i}) = 0$ *for* $i = 1, 2$, *and so* $\sigma(S_{\mu^1} \cup S_{\mu^2}) = 0$. *Therefore, similarly to the case of discrete measures with finite support, given any* $x \in \mathrm{supp}(\mu^1)$, $y \in \mathrm{supp}(\mu^2)$ *we have that the map* $\theta \mapsto \frac{p(x)q(y)}{P(\bar{x}^\theta)Q(\bar{y}^\theta)}$ *from* $\mathbb{S}^{d-1}$ *to* $\mathbb{R}$ *is equal to the constant function* $\theta \mapsto 1$ *up to a set of $\sigma$-measure $0$. This implies that the function* $\theta \mapsto u_\theta^{\mu^1, \mu^2}$ *is measurable. Finally, since* $|\gamma_\theta^{\mu^1, \mu^1}(\{(x, y)\})| \leq 1$ *for every* $(x, y)$, *we have that* $\bar{\gamma}^{\mu^1, \mu^2}$ *is well-defined.*

## B EQUIVALENCE WITH WEAK* CONVERGENCE

**Lemma B.1.** *Let* $\Omega \subset \mathbb{R}$ *be a compact set,* $\mu \in \mathcal{P}(\Omega)$ *and consider a sequence of probability measures* $(\mu_n)_{n \in \mathbb{N}}$ *defined in* $\Omega$ *such that* $\mu_n \overset{*}{\rightharpoonup} \mu$ *as* $n \to \infty$. *Then, for each* $\theta \in \mathbb{S}^{d-1}$, *we have that* $\theta_{\#}\mu_n \overset{*}{\rightharpoonup} \theta_{\#}\mu$ *as* $n \to \infty$.

*Proof.* Given $\theta \in \mathbb{S}^{d-1}$, notice that $\{\theta \cdot x : x \in \Omega\}$ is a 1-dimensional compact set, which contains the supports of $\theta_{\#}\mu$ and $(\theta_{\#}\mu_n)_{n \in \mathbb{N}}$. Thus, when dealing with the weak*-topology we can use continuous functions as test functions. Let $\varphi : \mathbb{R} \to \mathbb{R}$ be a continuous test function, then

$$\int_{\mathbb{R}} \varphi(u) d\theta_{\#}\mu_n(u) = \int_{\mathbb{R}^d} \varphi(\theta \cdot x) d\mu_n(x) \xrightarrow[n \to \infty]{} \int_{\mathbb{R}^d} \varphi(\theta \cdot x) d\mu(x) = \int_{\mathbb{R}} \varphi(u) d\theta_{\#}\mu(u)$$

since the composition $x \mapsto \theta \cdot x \mapsto \varphi(\theta \cdot x)$ is a continuous function from $\mathbb{R}^d$ to $\mathbb{R}$. $\qquad \square$

**Lemma B.2.** *Let $\Omega \subset \mathbb{R}$ be a compact set, $\mu^i \in \mathcal{P}(\Omega)$, $i = 1, 2$, and consider sequences of probability measures $(\mu_n^i)_{n\in\mathbb{N}}$ defined in $\Omega$ such that $\mu_n^i \overset{*}{\rightharpoonup} \mu^i$ as $n \to \infty$, for $i = 1, 2$. Given $\theta \in \mathbb{S}^{d-1}$, consider $\Lambda_\theta^{\mu^1,\mu^2}$ the unique optimal transport plan between $\theta_\# \mu^1$ and $\theta_\# \mu^2$, and for each $n \in \mathbb{N}$, consider $\Lambda_\theta^{\mu_n^1,\mu_n^2}$ the unique optimal transport plan between $\theta_\# \mu_n^1$ and $\theta_\# \mu_n^2$. Then $\Lambda_\theta^{\mu_n^1,\mu_n^2} \overset{*}{\rightharpoonup} \Lambda_\theta^{\mu^1,\mu^2}$.*

*Proof.* The proof is similar to that of Lemma A.8. From Lemma B.1, Proposition A.6, and uniqueness of optimal plans in one-dimension, there exists a subsequence $(\Lambda_\theta^{\mu_{n_k}^1,\mu_{n_k}^2})_{k\in\mathbb{N}}$ such that

$$\Lambda_\theta^{\mu_{n_k}^1,\mu_{n_k}^2} \rightharpoonup^* \Lambda_\theta^{\mu^1,\mu^2}.$$

Now, suppose that the original sequence $(\Lambda_\theta^{\mu_n^1,\mu_n^2})_{n\in\mathbb{N}}$ does not converge to $\Lambda_\theta^{\mu^1,\mu^2}$ (in the weak*-topology). Thus, there exists a continuous function $\varphi : \mathbb{R} \to \mathbb{R}$ such that for a given $\varepsilon > 0$ there exists a subsequence $(\Lambda_\theta^{\mu_{n_j}^1,\mu_{n_j}^2})_{j\in\mathbb{N}}$ with

$$\left| \int_\mathbb{R} \varphi(u)\, d\Lambda_\theta^{\mu_{n_j}^1,\mu_{n_j}^2}(u) - \int_\mathbb{R} \varphi(u)\, d\Lambda_\theta^{\mu^1,\mu^2}(u) \right| > \varepsilon \tag{27}$$

But again, from Lemma B.1, Proposition A.6, and uniqueness of optimal plans in one-dimension, we have that there exists a sub-subsequence such that

$$\int_\mathbb{R} \varphi(u)\, d\Lambda_\theta^{\mu_{n_{j_i}}^1,\mu_{n_{j_i}}^2}(u) \underset{i\to\infty}{\longrightarrow} \int_\mathbb{R} \varphi(u)\, d\Lambda_\theta^{\mu^1,\mu^2}(u),$$

contradicting (27). Since the contradiction is achieved from assuming that the whole sequence $(\Lambda_\theta^{\mu_n^1,\mu_n^2})_{n\in\mathbb{N}}$ does not converge to $\Lambda_\theta^{\mu^1,\mu^2}$, we have that, in fact, it does converge to $\Lambda_\theta^{\mu^1,\mu^2}$ in the weak*-topology. $\square$

**Lemma B.3.** *Consider probability measures supported in a compact set $\Omega \subset \mathbb{R}^d$ such that $\mu_n^1 \overset{*}{\rightharpoonup} \mu^1$, $\mu_n^2 \overset{*}{\rightharpoonup} \mu^2$. Let $\theta \in \mathbb{S}^{d-1}$, then the sequence of lifted plans $\{\gamma_\theta^{\mu_n^1,\mu_n^2}\}_{n\in\mathbb{N}}$ satisfies that there exists a subsequence such that $\gamma_\theta^{\mu_{n_k}^1,\mu_{n_k}^2} \overset{*}{\rightharpoonup} \gamma^*$ for*

$$\gamma^* \in \Gamma(\mu^1, \mu^2; \Lambda_\theta^{\mu^1,\mu^2}) := \{\gamma \in \Gamma(\mu^1, \mu^2) : (\theta \times \theta)_\# \gamma = \Lambda_\theta^{\mu^1,\mu^2}\},$$

*where $\theta \times \theta(x,y) := (\theta \cdot x, \theta \cdot y)$ for all $(x,y) \in \mathbb{R}^d \times \mathbb{R}^d$.*

*Proof.* Since $\gamma_\theta^{\mu_n^1,\mu_n^2} \in \Gamma(\mu_n^1, \mu_n^2)$, similar to the proof of Proposition A.6, by Banach-Alaoglu Theorem, there exists a subsequence $\gamma_\theta^{\mu_{n_k}^1,\mu_{n_k}^2}$, such that $\gamma_\theta^{\mu_{n_k}^1,\mu_{n_k}^2} \overset{*}{\rightharpoonup} \gamma^*$, for some $\gamma^* \in \mathcal{P}(\Omega \times \Omega)$. Again, as in the proof of Proposition A.6, it can be shown that $\gamma^* \in \Gamma(\mu^1, \mu^2)$.

In addition, we have

$$(\theta \times \theta)_\# \gamma_\theta^{\mu_{n_k}^1,\mu_{n_k}^2} \overset{*}{\rightharpoonup} (\theta \times \theta)_\# \gamma^* \quad \text{and} \quad (\theta \times \theta)_\# \gamma_\theta^{\mu_{n_k}^1,\mu_{n_k}^2} = \Lambda_\theta^{\mu_{n_k}^1,\mu_{n_k}^2} \overset{*}{\rightharpoonup} \Lambda_\theta^{\mu^1,\mu^2}$$

(where the first one follows by similar arguments to those in Lemma B.1 and the second one follows by definition of the lifted plans). By the uniqueness of the limit (for the weak*-convergence), we have $(\theta \times \theta)_\# \gamma^* = \Lambda_\theta^{\mu^1,\mu^2}$. Thus, $\gamma^* \in \Gamma(\mu^1, \mu^2; \Lambda_\theta^{\mu^1,\mu^2})$. $\square$

**Theorem B.4.** *Let $\sigma \ll Unif(\mathbb{S}^{d-1})$. Consider discrete probability measures measures $\mu^1 = \sum_{i=1}^\infty p_i \delta_{x_i}, \mu^2 = \sum_{j=1}^\infty q_j \delta_{y_j}$ in $\Omega$ supported on a compact set $\Omega \subset \mathbb{R}^d$. Consider sequences $(\mu_n^1)_{n\in\mathbb{N}}, (\mu_n^2)_{n\in\mathbb{N}}$ of discrete probability measures defined on $\Omega$ such that $\mu_n^1 \rightharpoonup^* \mu^1, \mu_n^2 \rightharpoonup^* \mu^2$. Then $\sigma$-a.s. we have that $\mathcal{D}_p(\mu_n^1, \mu_n^2; \theta) \to \mathcal{D}_p(\mu^1, \mu^2; \theta)$ as $n \to \infty$. Moreover, $\mathcal{D}_p(\mu_n^1, \mu_n^2) \to \mathcal{D}_p(\mu^1, \mu^2)$ as $n \to \infty$.*

*Proof.* Let us define the set

$$S(\mu^1, \mu^2) := \left\{ \theta \in \mathbb{R}^d : \Gamma(\mu^1, \mu^2; \Lambda_\theta^{\mu^1, \mu^2}) = \{\gamma_\theta^{\mu^1, \mu^2}\} \right\}.$$

Since we are considering discrete measures, notice that

$$\mathbb{S}^{d-1} \setminus S(\mu^1, \mu^2) \subseteq S_{\mu^1} \cup S_{\mu^2},$$

where $S_{\mu^1} = \{\theta \in \mathbb{S}^{d-1} : \theta \cdot x_m = \theta \cdot x_{m'} \text{ for some pair } m \neq m'\}$ and $S_{\mu^2} = \{\theta \in \mathbb{S}^{d-1} : \theta \cdot y_n = \theta \cdot y_{n'} \text{ for some pair } n \neq n'\}$.

By Lemma A.13, we have $\sigma(\mathbb{S}^{d-1} \setminus S(\mu^1, \mu^2)) \leq \sigma(S_{\mu^1} \cup S_{\mu^2}) = 0$. Thus,

$$\sigma(S(\mu^1, \mu^2)) = 1.$$

Let $\theta \in S(\mu^1, \mu^2)$, and consider the lifted plans $\gamma_\theta^{\mu^1, \mu^2}$ and $\gamma_\theta^{\mu_n^1, \mu_n^2}$. By Lemma B.3, there exists a subsequence of $(\gamma_\theta^{\mu_n^1, \mu_n^2})_{n \in \mathbb{N}}$ such that

$$\gamma_\theta^{\mu_{n_k}^1, \mu_{n_k}^2} \overset{*}{\rightharpoonup} \gamma^* \in \Gamma(\mu^1, \mu^2; \Lambda_\theta^{\mu^1, \mu^2}).$$

Since $\theta \in S(\mu^1, \mu^2)$, we have that $\Gamma(\mu^1, \mu^2; \Lambda_\theta^{\mu^1, \mu^2})$ contains only one element, which is $\gamma_\theta^{\mu^1, \mu^2}$. Hence,

$$\gamma^* = \gamma_\theta^{\mu^1, \mu^2}.$$

Moreover, by uniqueness of weak convergence, proceeding similarly as in the proof of Lemma B.2, we have that the whole sequence $(\gamma_\theta^{\mu_n^1, \mu_n^2})_{n \in \mathbb{N}}$ converges to $\gamma_\theta^{\mu^1, \mu^2}$ in the weak*-topology.

Therefore, by definition of weak*-convergence for measures supported in a compact set (in our case, $\Omega \times \Omega \subset \mathbb{R}^d \times \mathbb{R}^d$), since $(x, y) \mapsto \|x - y\|^p$ is a continuous function, we have

$$\begin{aligned}
\lim_{n \to \infty} \mathcal{D}_p(\mu_n^1, \mu_n^2; \theta)^p &= \lim_{n \to \infty} \int_{\Omega^2} \|x - y\|^p d\gamma_\theta^{\mu_n^1, \mu_n^2}(x, y) \\
&= \int_{\Omega^2} \|x - y\|^p d\gamma_\theta^{\mu^1, \mu^2}(x, y) \\
&= \mathcal{D}_p(\mu^1, \mu^2; \theta)^p
\end{aligned}$$

Combining this with the fact that $\sigma(S(\mu^1, \mu^2)) = 1$ and that $(\mathcal{D}_p(\mu_n^1, \mu_n^2; \theta)^p)_{n \in \mathbb{N}}$ is bounded, that is, $|\mathcal{D}_p(\mu_n^1, \mu_n^2; \theta)^p| \leq \max_{(x,y) \in \Omega \times \Omega} \|x - y\|^p$, by Dominated Lebesgue Theorem we obtain

$$\begin{aligned}
\lim_{n \to \infty} \mathcal{D}_p(\mu_n^1, \mu_n^2)^p &= \lim_{n \to \infty} \int_{\mathbb{S}^{d-1}} \mathcal{D}_p(\mu_n^1, \mu_n^2; \theta)^p d\sigma(\theta) \\
&= \int_{\mathbb{S}^{d-1}} \lim_{n \to \infty} \mathcal{D}_p(\mu_n^1, \mu_n^2; \theta)^p d\sigma(\theta) \\
&= \int_{\mathbb{S}^{d-1}} \mathcal{D}_p(\mu^1, \mu^2; \theta)^p d\sigma(\theta) \\
&= \mathcal{D}_p(\mu^1, \mu^2)^p
\end{aligned}$$

$\square$

**Corollary B.5.** *Let $\mu, \mu_n \in \mathcal{P}(\Omega)$, where $\Omega \subset \mathbb{R}^d$ is compact, be of the form $\mu = \sum_{x \in \mathbb{R}^d} p(x) \delta_x$, $\mu_n = \sum_{x \in \mathbb{R}^d} p_n(x) \delta_x$ where $p(x)$ and $p_n(x)$ are 0 at all but countably many $x \in \mathbb{R}^d$. Assume $\sigma \ll Unif(\mathbb{S}^{d-1})$. Then, $\mathcal{D}_p(\mu_n, \mu) \to 0$ if and only if $\mu_n \overset{*}{\rightharpoonup} \mu$.*

*Proof.* If $\mathcal{D}_p(\mu_n, \mu) \underset{n \to \infty}{\longrightarrow} 0$ then, by Remark 2.8, $W_p(\mu_n, \mu) \underset{n \to \infty}{\longrightarrow} 0$, hence $\mu_n \underset{n \to \infty}{\overset{*}{\rightharpoonup}} \mu$.

The converse is a Corollary of Theorem B.4

$\square$

## C  Computational efficiency

To deduce the computational complexity of the proposed method, we consider, for simplicity, two finite discrete probability measures on $\mathbb{R}^d$, $d > 1$, concentrated at $N$ particles, $\mu = \sum_{i=1}^{N} p_i \delta_{x_i}$ and $\nu = \sum_{j=1}^{N} q_j \delta_{y_j}$. Consider $L$ slices or unit vectors $\{\theta_l\}_{l=1}^{L}$ in $\mathbb{S}^{d-1}$ We assume that no overlap occurs when projecting along different directions (as previously noted, this is almost surely the case as proven in Lemma A.13). The following is the analysis of the computational complexity of the proposed EST method:

- For each one of the $L$ directions, $\{\theta_l\}_{l=1}^{L}$, we have to project the locations $\{x_i\}_{i=1}^{N}$, $\{y_j\}_{j=1}^{N}$ to obtain the new projected measures concentrated at $\{\theta_l \cdot x_i\}_{i=1}^{N}$ and $\{\theta_l \cdot y_j\}_{j=1}^{N}$, respectively. This requires $\mathcal{O}(LdN)$ operations.

- To compute the one-dimensional plan $\Lambda_{\theta_l}^{\mu,\nu}$, for each $L \le \ell \le L$, we essentially need to solve a one-dimensional optimal transport problem (i.e., a sorting problem), which is of order $\mathcal{O}(N \log(N))$. Thus, when considering all the slices, this step is of order $\mathcal{O}(LN \log(N))$.

- The lifting process that gives rise to $\gamma_{\theta_l}^{\mu,\nu}$ does not require additional operations to be taken into account (it is performed as an assignment or correspondence).

- The plan $\bar{\gamma}^{\mu,\nu}$ can be represented as an $N \times N$ matrix. The $(i,j)$-entry is given by $\sum_{l=1}^{L} \gamma_{\theta_l}^{\mu,\nu}(\{(x_i, y_j)\})$, requiring $L$ operations. Thus, the complexity of this step is $\mathcal{O}(LN^2)$.

- Finally, once we have $\bar{\gamma}^{\mu,\nu}$, for computing the EST-distance we require another $\mathcal{O}(N^2 d)$ operations.

As a conclusion, the complexity of computing the plan $\bar{\gamma}^{\mu,\nu}$ is $\mathcal{O}(L(Nd + N\log(N) + N^2))$, or simply, $\mathcal{O}(LN^2 + LNd)$, and that of the EST-distance is $\mathcal{O}((L+d)N^2)$. As it is generally the case, $N$ is much larger than $d$, so the **computational complexity** of both the EST-plan and the EST-distance is of order

$$\mathcal{O}((L+d)N^2).$$

We recall that the complexity of using the linear programming approach for solving the classical optimal transport problem between $\mu$ and $\nu$ is of order $\mathcal{O}(N^3 \log(N))$. Besides, for the entropic regularized version solved by using iterative algorithms like Sinkhorn's algorithm, the complexity is of order $\mathcal{O}(N^2 \log(N)/\lambda^2)$, where $\lambda$ denotes the regularization parameter (see Dvurechensky et al. (2018); Altschuler et al. (2017)).

In Figures 8 and 9, we report the wall-clock times of the Sinkhorn algorithm, used for computing the entropic regularized version of OT (for different values of the regularization parameter $\lambda$), compared to the proposed EST distance (calculated with different numbers of slices $L$) between two $N$-sized empirical distributions. The only difference between the two figures is the scale used for the horizontal axis, which has been adjusted for visualization purposes.

**Remark C.1.** *It is important to note that the computation of EST method can be parallelized over the number of slices $L$. Currently, our code does not implement parallelization, but incorporating it would significantly speed up the computation of the EST plan and distance.*

## D  Additional Experimental Results

To further demonstrate the efficiency of our proposed method for the classification task, we consider the widely used benchmark dataset in 3D computer vision and geometric deep learning, ModelNet40 (Wu et al., 2015). This dataset consists of objects represented as 3D point clouds, with 2048 points per object. It contains 40 different object categories. The training set comprises 9,840 samples, and the test set includes 2,468 samples.

In Table 1, we compute the accuracy of the 1NN classifier using (1) the classical optimal transport (OT) approach calculated with linear programming (LP), (2) its entropic regularized version for different regularization parameters $\lambda$ using the Sinkhorn algorithm, and (3) our proposed EST method with different numbers of slices ($L$) or unit vectors in $\mathbb{S}^2$.

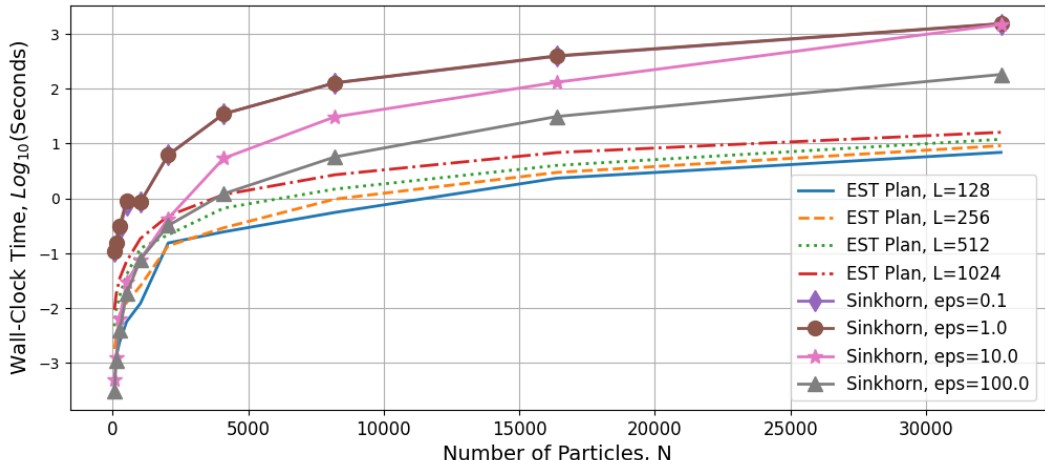

Figure 8: Wall-clock time between Sinhkorn algorithm applied for computing entropic OT for different values of the regularization parameter $\lambda$ and the proposed EST method calculated with different numbers $L$ of slices. We compare the differences in time as the size of $N$ the empirical measures increases.

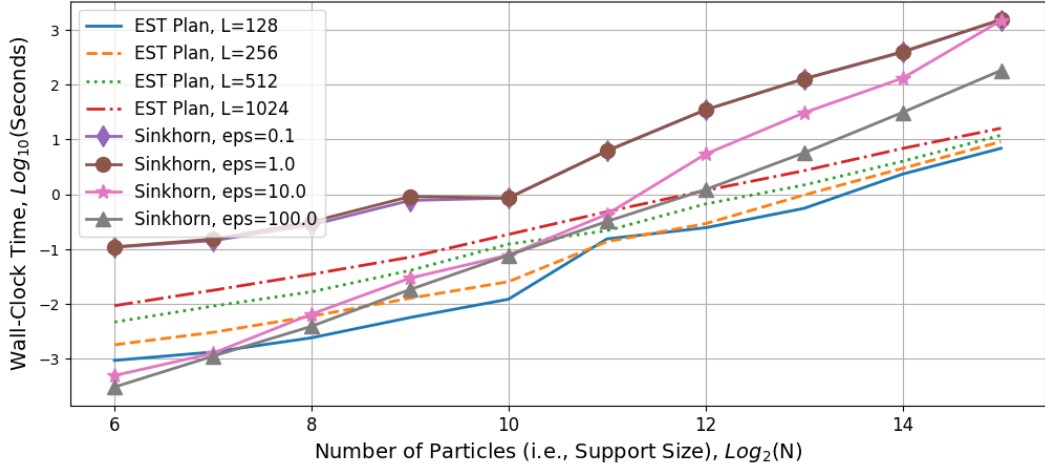

Figure 9: $\log_2 - \log_{10}$ wall-clock time plot between Sinhkorn algorithm applied for computing entropic OT for different values of the regularization parameter $\lambda$ and the proposed EST method calculated with different numbers $L$ of slices. We compare the differences in time as the size of $N$ the empirical measures increases.

Of importance, we used the linearized version of these three methods, as described in Subsection 3.6. Direct computation of the 1NN classifier purely based on OT, its entropic regularized version, and the EST distance would require $2,048 \times 9,840 = 20,152,320$ ("number of testing samples $\times$ number of training samples") pairwise comparisons for each of the three approaches. Instead, we compute $2,048 + 9,840 = 11,888$ ("number of testing samples $+$ number of training samples") transport plans (with the three different methods) and then use a traditional classifier with Euclidean distance on the approximated Monge maps given by (17).

We recall that the linearization technique is based on pivoting with a reference measure $\mu_0$ to reduce the number of pairwise comparisons. Specifically, it involves computing transport plans (using the different transportation approaches: OT, entropic regularization, or EST) between $\mu_0$ and the target samples (training and testing point clouds), applying the barycentric projection (16) to obtain an embedding (17) of all the training and testing samples, and using a classifier in the embedding space.

| 1NN Classification | Sinkhorn $\lambda = 10$ | Sinkhorn $\lambda = 1$ | ESP $L = 128$ | ESP $L = 1024$ | OT (LP) |
|---|---|---|---|---|---|
| Accuracy $\uparrow$ | 65.96% | 78.93% | 77.30% | 79.45% | 82.09% |
| Time per distance (Sec) $\downarrow$ | 0.423 | 0.594 | 0.185 | 0.368 | 0.883 |

Table 1: Accuracy and time comparison of 1NN classifier implemented after the embedding.

In these experiments, for all methods, the reference measure $\mu_0$ is chosen as the uniform measure on the cube $[-1, 1]^3$, and we use a 1NN classifier with Euclidean distance in the embedding space.

# E   RELATIONS BETWEEN THE AVERAGING MEASURE $\sigma_\tau$ AND THE SLICING DISTRIBUTION IN ENERGY-BASED SLICED WASSERSTEIN DISTANCE

We draw an analogy between our proposed averaging measure $\sigma_\tau$ defined in 14 and the slicing distribution in the energy-based Sliced Wasserstein distance Nguyen & Ho (2024). Under the assumption that a higher value of 1-dimensional Wasserstein distance $W_p^p(\theta_{\#}\mu, \theta_{\#}\nu)$ will give a better projecting direction $\theta$, Nguyen & Ho (2024) propose to build on the Sliced Wasserstein distance and define a slicing distribution supported on $\mathbb{S}^{d-1}$ as

$$\sigma_{\mu,\nu}(\theta; f, p) := \frac{f(W_p^p(\theta_{\#}\mu, \theta_{\#}\nu))}{\int_{\mathbb{S}^{d-1}} f(W_p^p(\theta'_{\#}\mu, \theta'_{\#}\nu)) d\theta'}$$

for $\mu, \nu \in \mathcal{P}_p(\mathbb{R}^d)$, where $f : [0, \infty) \to \Theta \subset (0, \infty)$ is a monotonically increasing energy function. When $f$ is the exponential function $f_e(x) = e^x$, the slicing distribution becomes

$$d\sigma_{\mu,\nu}(\theta; f, p) = \frac{e^{W_p^p(\theta_{\#}\mu, \theta_{\#}\nu)}}{\int_{\mathbb{S}^{d-1}} e^{W_p^p(\theta'_{\#}\mu, \theta'_{\#}\nu)} d\theta'} d\theta \tag{28}$$

Both our proposed $\sigma_\tau$ in Equation 14 and $\sigma_{\mu,\nu}(\theta; f, p)$ in Equation 28 employ the exponential function to reflect the optimality of slices in the resulting distances, with Wasserstein distance serving as the benchmark. However, it's important to note a key distinction: contrary to the Sliced Wasserstein distance as a lower bound of Wasserstein distance, the Expected Sliced Transport distance is an upper bound, as for all $\theta \in \mathbb{S}^{d-1}$,

$$W_p(\theta_{\#}\mu, \theta_{\#}\nu) \leq W_p(\mu, \nu) \leq \mathcal{D}_p(\mu, \nu; \theta).$$

Thus, the analogous energy function in $\sigma_\tau$ is $g(x) = e^{-\tau x}$ (with $\tau \geq 0$) which is monotonically decreasing, based on the assumption that a better slicing direction $\theta$ will result in a lower $\mathcal{D}_p(\mu, \nu; \theta)$.

