# OpenReview forum: "Expected Sliced Transport Plans"
_ICLR.cc/2025/Conference — ICLR 2025 Poster_

### Official Review · Reviewer_fUu9 · 2024-10-31

**Soundness:** 4
**Presentation:** 3
**Contribution:** 3
**Rating:** 8
**Confidence:** 4

**Summary:**

The paper introduces expected sliced optimal transport plans with four contributions: computational efficiency, a distance for discrete probability measures, theory and experiments demonstrating equivalence with Wasserstein distances, and use case demonstrations.

**Strengths:**

- The topic, approach, and technical exposition were all excellent.
- The theoretical results were interesting and clear.
- The empirical results were interesting including the synthetic demonstrations and real world use cases.

**Weaknesses:**

- I found it hard to understand what the paper was aiming for from the introduction. The intent of the paper came into sharper focus for me well after the start of the technical section. Currently the introduction is relatively abstract in motivating the approach. I wonder whether the authors wouldn't be better served by more concrete motivations earlier. Often this is done by having a figure to illustrate the approach. Maybe figure 1 could be this if it were earlier? However, that figure would benefit from more explanation.
- The introduction focuses on computational efficiency; however, I don't see any empirical results demonstrating this point. Because the datasets being used here aren't huge by machine learning standards, it seems important to be precise about what is meant by this claim.

Minor confusions / questions that arise while I was reading:
- "Traditional models scale cubically" What methods are being invoked here?
- "However, the number of iterations required for convergence typically increases as the regularization parameter decreases, which can offset the computational benefits of these methods." I don't understand this critique.
- Around line 46: I see the citation to Cuturi there, but the discussion is inadequate: "Various approaches have been 046 developed to address this challenge, including entropic regularization (Cuturi, 2013),"
- Why are we introducing transport maps in equation 5?

**Questions:**

Please see above.

---

> ### Author Response · Authors · 2024-11-23
> **Response to Reviewer fUu9**
>
> Thank you for your thorough evaluation of our paper and for your time and consideration. Below, we provide our responses and clarifications.
>
> ### Weaknesses:
>
> > I found it hard to understand what the paper was aiming for from the introduction. The intent of the paper came into sharper focus for me well after the start of the technical section. Currently, the introduction is relatively abstract in motivating the approach. I wonder whether the authors wouldn't be better served by more concrete motivations earlier. Often this is done by having a figure to illustrate the approach. Maybe figure 1 could be this if it were earlier? However, that figure would benefit from more explanation.
>
> **Answer:** We thank the reviewer for this valuable feedback. We took several actions to address this concern. First, to provide a summary of our new technique in the Introduction, we have rewritten our first bullet point contribution as follows: "Introducing a computationally efficient transport plan between discrete probability measures, the \textbf{Expected Sliced Transport (EST) plan}.  Motivated by the first question highlighted above, we construct this transport plan as the average of transport plans computed via a lifting scheme involving one-dimensional sliced transport plans. (See Definition 2.6 below.)"
>
> Second, we moved Figure 1 to the second page of our paper and updated the caption to clearly explain the proposed idea and how to interpret this Figure. In particular, we updated the caption as follows:
>
> "Our goal in this paper is to construct transportation plans between $d$-dimensional measures ($d > 1$) using their one-dimensional marginals, or slices. For two measures $\mu^1$ (represented by green circles) and $\mu^2$ (represented by blue circles), we first derive a one-dimensional transport plan $\Lambda_\theta^{\mu^1, \mu^2}$ from their slices along a unit vector $\theta$ (indicated by triangles). This one-dimensional transport plan is then lifted back to the original $d$-dimensional space to produce a transport plan between the measures, $\gamma_\theta^{\mu^1, \mu^2}$. Finally, these transport plans are aggregated over all directions $\theta \in \mathbb{S}^{d-1}$. In (a), the measures $\mu^1$ and $\mu^2$ are uniform, with their masses non-overlapping when projected in the direction of $\theta$. In (b), $\mu^1$ and $\mu^2$ are non-uniform, resulting in overlapping masses when projected along $\theta$. For further details, see Remark A.1 in Appendix A.1."
>
> Given the page limit constraints, we hope these revisions have enhanced the clarity of our paper.

---

> > ### Author Response · Authors · 2024-11-23
> >
> > > The introduction focuses on computational efficiency; however, I don't see any empirical results demonstrating this point. Because the datasets being used here aren't huge by machine learning standards, it seems important to be precise about what is meant by this claim.
> >
> > **Answer:** We agree with the reviewer that this is a key point and that we need to emphasize it more clearly. For this reason, we have included the new Subsection 3.1 and Appendix C on the computational efficiency of our proposed method. Below is a brief overview of these arguments.
> >
> > For the following analysis, let us consider two finite discrete probability measures on $\mathbb R^d$, $d>1$, concentrated at $n$ particles ($\mu=\sum_{i=1}^n p_i\delta_{x_i}$, $\nu=\sum_{j=1}^n q_j\delta_{y_j}$).  The complexity of using the linear programming approach for solving the optimal transport problem between them is of order $\mathcal{O}(n^3\log(n))$.  When using entropic regularization, the problem becomes easier to solve using iterative algorithms like Sinkhorn's algorithm, having complexity of order $\mathcal{O}(n^2 \log(n)/\lambda^2)$, where $\lambda$ is the regularization parameter
> > ([Dvurechensky et al., 2018](https://arxiv.org/pdf/1802.04367), [Altschuler et al., 2018](https://arxiv.org/pdf/1705.09634)), but introducing an approximation to the original problem.
> >
> > When dealing with measures in the real line, by using sorting algorithms, we reach a complexity of order $\mathcal{O}(n\log(n))$. The proposed EST method has following computational complexity when considering $L$ slices or unit vectors $\\{\theta_l\\}_{l=1}^L$ in $\mathbb{S}^{d-1}$:
> >
> > - For each one of the $L$ directions ($\\{ \theta_l \\}_{l=1}^L$), the slicing operation involves  $\mathcal{O}(Ldn)$ operations to obtain $\theta_l \cdot x_i $ and $\theta_l \cdot y_j $ for $i,j \in [1,n]$.
> >
> > - To compute the 1D plan $\Lambda_{\theta_l}^{\mu,\nu}$, for each $L\leq \ell\leq L$, we essentially need to solve a one-dimensional optimal transport problem (sorting problem), which is of order $\mathcal O(n\log(n))$. Thus, when considering all the slices, this step is of order $\mathcal O(Ln\log n)$.
> > - The lifting process that gives rise to $\gamma_{\theta_l}^{\mu,\nu}$ does not requires operations to be taken into account.
> > - The plan $\bar\gamma^{\mu,\nu}$ can be represented as an $n\times n$ matrix. The $(i,j)$-entry is given by $\sum_{l=1}^L\gamma_{\theta_l}^{\mu,\nu}(\{(x_i,y_j)\})$, requiring $L$ operations. Thus, the complexity of this step is $\mathcal O(L n^2)$.
> > - Finally, once we have $\bar\gamma^{\mu,\nu}$,  for computing the EST-distance we require another $\mathcal O(n^2d)$ operations.
> >
> > Thus, the complexity of computing the plan $\bar\gamma^{\mu,\nu}$ is $\mathcal O(L(nd+n\log n+n^2))$, or simply, $\mathcal O(Ln^2+Lnd)$, and that of the EST-distance is $\mathcal{O}((L+d)n^2)$. As it is generally the case, $n$ is much larger than $d$, so the computational complexity of both the EST-plan and the EST-distance is of order $\mathcal{O}((L+d)n^2)$.
> >
> >
> > Besides, in the following Figures ([Fig.1](https://postimg.cc/kDvL4ySn) - [Fig.2](https://postimg.cc/N5GSrr8R)) we report the wall-clock times of the Sinkhorn algorithm, used for computing the entropic regularized version of OT (for different values of the regularization parameter $\lambda$), compared to the proposed EST distance (calculated with different numbers of slices $L$) between two $N$-sized empirical distributions. The only difference between the two figures is the scale used for the horizontal axis, which has been adjusted for visualization purposes.
> >
> > We have added this discussion and figures in the new Subsection 3.1 and in the Supplementary Material (Appendix C).
> >
> > > "Traditional models scale cubically" What methods are being invoked here?
> >
> > **Answer:**
> > We thank the reviewer for this comment and we will clarify this issue in the revised version as follows: "Traditional models scale cubicallyTraditional OT solvers for discrete measures typically scale cubically with the number of samples $n$ (i.e., the support size)  (Kolouri et al., 2017, Peyre and Cuturi, 2019). Precisely, when using linear programming, the computational complexity is $\mathcal{O}(n^3\log(n))$."
> >
> > > "However, the number of iterations required for convergence typically increases as the regularization parameter decreases, which can offset the computational benefits of these methods." I don't understand this critique.
> >
> > **Answer:** This is a good point. Precisely, the complexity of Sinkhorn method is of order $\mathcal{O}(n^2 \ln(n)/\lambda^2)$, where $\lambda$ is the regularization parameter in the regularized version of OT
> > [Dvurechensky et al., 2018](https://arxiv.org/pdf/1802.04367), [Altschuler et al., 2018](https://arxiv.org/pdf/1705.09634). Thus, $\lambda$ decreases, and the number of iterations and hence the corresponding operations increases.
> >
> > We thank the reviewer for their comment. We have included this clarification in the new version of the paper.

---

> > > ### Author Response · Authors · 2024-11-23
> > >
> > > > Around line 46: I see the citation to Cuturi there, but the discussion is inadequate: "Various approaches have been 046 developed to address this challenge, including entropic regularization (Cuturi, 2013),"
> > >
> > > **Answer:** We have revised such line and write it as: "Various approaches have been developed to address this challenge, including the seminal work of Cuturi, 2013, which introduces entropic regularization, leveraging the Sinkhorn algorithm to compute OT efficiently with quadratic time complexity; ... "
> > >
> > > > Why are we introducing transport maps in equation 5?
> > >
> > > **Answer:** The one-dimensional transport maps are introduced at the beginning of the paper to motivate the subsequent construction, specifically the construction of the lifted plans $\gamma_\theta$. Moreover, Remark 2.9 revisits Equation (5) to establish the connection with prior works (in particular, Rowland et al. AISTATS 2019).
> > >
> > > Thank you once again for upholding high standards in your reviews. Your constructive feedback has helped us significantly improve our paper.

---

> > > > ### Comment · Reviewer_fUu9 · 2024-11-25
> > > > **Thank you for the the thoughtful response**
> > > >
> > > > Thanks to the authors for their thoughtful responses to my and the other reviewers' questions. I am quite satisfied by the clarifications and changes to the paper. I have increased my score accordingly.

---

### Official Review · Reviewer_Uqt2 · 2024-11-04

**Soundness:** 3
**Presentation:** 3
**Contribution:** 2
**Rating:** 6
**Confidence:** 3

**Summary:**

The authors propose to use a "lifting" technique to recover a transportation plan from sliced transportation maps. The lifted transportation plan applies to labels of discrete measures, They proved that the plan marginalizes to the original measures. Furthermore, the authors compute the expectation of the lifted transportation plan and plug the expectation into the original transportation problem. They finally proved that this transportation cost induced by their expected lift plan defines a metric. The authors later show empirical evidence for convergence of their new plan and distance with toy data and their metric properties with a MNIST variant.

**Strengths:**

+Paper is clear.

+The lifting technique is a good fit for translating the sliced OT problem back to the OT problem on the original measures.

+The authors' proposed contributions are well demonstrated in the paper, with the exception of (4) since their results are preliminary.

**Weaknesses:**

-One thing that I expected from the paper but didn't find is purpose of answering the two key questions.

First question "Can a transportation plan be constructed between two probability measures ...". Also in line 61, "limiting their applicability to problems that require explicit coupling between measures." What problems require explicit coupling between measures? Does this paper solve those problems?

Second question, "can this plan be used to define a metric between the measures". Maybe the interpolation experiment shows a little bit of the usage of the metric properties. Other than that, from reading this paper, I don't see the reason of studying the metric properties.

-The empirical results are weak because the dataset is too simple to prove its usefulness in real applications. They're at most for proof of concept.

**Questions:**

In 144: "As a result, it is more convenient to work with lifted transport plans."
Is plan the emphasis of the sentence? Instead of maps? And the difference between, as implied in this paper, is that maps apply to the support and plans apply to labels?


Please read proof the paper, to fix minor errors like "on the -(the) labels" in line 14, "the weights of +(the) optimal transport", and "the reason why we make this choice -(if) +(is)" in line 211

---

> ### Author Response · Authors · 2024-11-23
> **Response to Reviewer Uqt2**
>
> We sincerely thank the reviewer for their thoughtful feedback, time, and effort. We have revised the paper to address the reviewer's constructive suggestions. Below please find our responses.
>
> ### Strengths:
>
> > The authors' proposed contributions are well demonstrated in the paper, with the exception of (4) since their results are preliminary.
>
> **Answer:** We thank the reviewer for the following comment: "The authors' proposed contributions are well demonstrated in the paper, with the exception of (4) since their results are preliminary."
> In response to this, we would like to clarify our contribution listed as (4). In the original manuscript, it was stated as: "Demonstrating the performance of the proposed distance and the transportation plan in diverse applications, namely interpolation and classification." For clarity, we have revised this in the new version to: "Illustrating the potential applicability of the proposed distance and transport plan, with a focus on interpolation and classification tasks."
>
> ### Strengths:
>
> +The authors' proposed contributions are well demonstrated in the paper, with the exception of (4) since their results are preliminary.
>
> **Answer:** We thank the reviewer for the following comment: "The authors' proposed contributions are well demonstrated in the paper, with the exception of (4) since their results are preliminary."
> In response to this, we would like to clarify our contribution listed as (4). In the original manuscript, it was stated as: "Demonstrating the performance of the proposed distance and the transportation plan in diverse applications, namely interpolation and classification." For clarity, we have revised this in the new version to: "Illustrating the potential applicability of the proposed distance and transport plan, with a focus on interpolation and classification tasks."
>
> ### Weaknesses:
> > One thing that I expected from the paper but didn't find is purpose of answering the two key questions.
> >   1) First question Can a transportation plan be constructed between two probability measures ...".
>
> **Answer:** The first question stated in our paper reads as: "Can a transport plan between two probability measures on $\mathbb{R}^d$ be constructed using the sliced transport framework?"
>
> In this regard, we addressed this question in Definition 2.4, where we constructed a transport plan, denoted by $\bar{\gamma}^{\mu^1, \mu^2}$, between two probability measures $\mu^1$ and $\mu^2$. This plan is defined as the average of transport plans $\gamma_\theta^{\mu^1, \mu^2}$, which are computed through a lifting scheme involving one-dimensional sliced transport plans $\Lambda_\theta^{\mu^1, \mu^2}$.
>
> We thank the reviewer for pointing out that this was not clearly emphasized in the manuscript. To address this, we will revise the statement of our first contribution. Instead of:
>
> *"Introducing a computationally efficient transport plan between discrete probability measures, the Expected Sliced Transport plan."*
>
> We now propose:
>
> *"Introducing a computationally efficient transport plan between discrete probability measures, the Expected Sliced Transport plan. Motivated by the first question highlighted above, we construct this transport plan as the average of transport plans computed via a lifting scheme involving one-dimensional sliced transport plans. (See Definition 2.4 below.)"*
>
> We believe this revision will clarify how the paper addresses the question and make our contribution more explicit.
>
> > 2) Also in line 61, "limiting their applicability to problems that require explicit coupling between measures." What problems require explicit coupling between measures? Does this paper solve those problems?
>
> **Answer:** As an example, **interpolation** between two distributions of masses can be performed using the explicit coupling between them. In Section 3.3, we provide such interpolations between two probability measures $\mu$ and $\nu$ via $((1-t)x + ty)_{\\#} \gamma$. We used our proposed coupling $\bar \gamma$ and compared its performance with other standard transportation plans (see Figure 4).
>
> In Section 3.5, we applied our proposed transport plan to point cloud classification by interpreting point clouds as probability measures. Using the Linear Optimal Transport (LOT) framework, we introduced a reference measure $\mu_0$ and computed EST-transportation plans $\bar{\gamma}^{\mu_0, \mu_k}$ for all target point clouds $\mu_k$. Logistic regression was performed after barycentric projection (Eq. 16). Figure 6 compares our method with the classical LOT framework and "LOT + entropic OT," which uses entropic regularized transport plans followed by barycentric projection.
>
> These are only two simple applications that use a coupling between two distributions, but the scope of applications is not limited to these.

---

> > ### Author Response · Authors · 2024-11-23
> >
> > > 3) Second question, "can this plan be used to define a metric between the measures". Maybe the interpolation experiment shows a little bit of the usage of the metric properties. Other than that, from reading this paper, I don't see the reason of studying the metric properties.
> >
> > **Answer:** The second question highlighted in the Introduction of the paper reads: "can the resulting transportation plan be used to define a metric between the two probability measures?"
> >
> > The definition of our proposed discrepancy is provided in Definition 2.6, and its metric properties are outlined in Theorem 2.10. We have now made this explicit in our second bullet point contribution.
> >
> > The importance of having a metric in mathematics and its applications lies in the structure and rigor it provides for comparing elements—in our case, comparing probability measures in $\mathbb{R}^n$. A metric offers a precise and consistent way to measure the "distance" between two elements in a space.
> >
> > From a theoretical standpoint, defining a metric endow the probability space with a topological structure. This structure allows us to rigorously work with foundational concepts such as convergence and continuity, which are essential in both mathematical theory and practical applications.
> >
> > From a machine learning perspective, pairwise distances in a dataset are crucial for tasks such as manifold learning, constructing graph structures, dimensionality reduction, and designing kernel methods (e.g., kernel Support Vector Machines). A well-defined metric ensures that these algorithms are consistent and meaningful, which is essential for reliable optimization and learning.
> >
> > > The empirical results are weak because the dataset is too simple to prove its usefulness in real applications. They're at most for proof of concept.
> >
> > **Answer:** We agree with the reviewer that our experiments are conducted on simple datasets. Our goal was to keep the paper's message clear and focused on our theoretical findings, avoiding additional complexities that more intricate datasets might introduce. We also agree with the reviewer that including experiments with more complex data would greatly enhance the paper. To address this, we have added Appendix D, where we further demonstrate the efficiency of our proposed method for the classification task using ModelNet40, a widely used benchmark dataset in 3D computer vision and geometric deep learning. In [Figure](https://postimg.cc/mhbNBxGg), we compare the accuracy and computation time required for performing a 1-NN classifier using distance information provided by three different methods: OT, entropic regularization via the Sinkhorn algorithm, and the proposed EST method.
> >
> > Besides, as mentioned earlier in this rebuttal, we have revised our last listed contribution for clarity, and the updated version reads as follows: "Illustrating the potential applicability of the proposed distance and transport plan, with a focus on interpolation and classification tasks." Using the wording "potential applicability," we acknowledge that our experiments serve as proof of concept.
> >
> > On another note, the experiments presented in Section 3.4, which demonstrate the equivalence between the topology induced by our new metric and the topology given by the Wasserstein distance (and, consequently, the weak* topology), are rigorously proven in the appendix for discrete probability measures with finite or compact support.
> >
> > As stated in our paper, our ultimate goal is for the theoretical insights and experimental results presented here to inspire the development of efficient transport-based algorithms in machine learning and beyond.
> >
> > ### Questions:
> >
> > > (1) In 144: "As a result, it is more convenient to work with lifted transport plans." Is plan the emphasis of the sentence? Instead of maps? And the difference between, as implied in this paper, is that maps apply to the support and plans apply to labels?
> >
> > **Answer:**  We thank the reviewer for their thorough reading.
> >
> > The word "convenient" should be replaced by "appropriate". We will address this is the revised manuscript.
> >
> > Our emphasis in this sentence is that using transport plans is more appropriate because transport maps may not always be well-defined. Specifically, transport maps might not be defined on the support of the reference measure $\mu^1$. (However, as they are expressed in terms of permutations $\tau_\theta$ and $\zeta_\theta$ in the symmetric group, which operate on labels $\{1, \dots, N\}$ they can be defined on maps on those labels $\{1, \dots, N\}\mapsto \{1, \dots, N\}$).
> >
> > Intead, the lifted transport plans $\gamma_\theta^{\mu^1, \mu^2}$ are measures supported on $\{(x_i, y_j)\}$, where $\{x_i\}$ is the support of $\mu^1$ and $\{y_j\}$ is the support of $\mu^2$. Additionally, these lifted transport plans can be defined for measures supported on differing numbers of points, which transport maps alone cannot always accommodate.

---

> > > ### Author Response · Authors · 2024-11-23
> > >
> > > > (2) Please read proof the paper, to fix minor errors like "on the -(the) labels" in line 14, "the weights of +(the) optimal transport", and "the reason why we make this choice -(if) +(is)" in line 211.
> > >
> > > **Answer:** We thank the reviewer for thoroughly reading the paper. We have corrected all the identified typos in the revised manuscript.
> > >
> > > Thank you once again for upholding high standards in your reviews.

---

> ### Comment · Reviewer_Uqt2 · 2024-11-24
>
> Just a minor point. I would reverse the orders of the rows in Figure 4, so that rows with the best quality are closer to the original OT interpolation.

---

> ### Comment · Reviewer_Uqt2 · 2024-11-24
>
> I agree with "A metric offers a precise and consistent way to measure the "distance" between two elements in a space." My point was that the purpose of proving the metric properties in the paper was not clear. Why is it important that "the resulting transportation plan be used to define a metric between the two probability measures"? What if it's a pseudo metric? What applications of the plan will not hold if it doesn't define a metric? I understand the benefits of a true metric but I didn't find the benefits demonstrated in the paper.

---

> ### Comment · Reviewer_Uqt2 · 2024-11-24
>
> But we don't necessarily need explicit couplings to interpolate between the marginals, do we? We could solve a barycenter problem to get the interpolated measures. With the explicit couplings, yes, it's more convenient but I don't think it's required. So back to my original point, I was not arguing about the questions themselves. I was asking about the purpose of asking those questions. Given prior work on Sliced OT and OT in general, what is the purpose of an explicit plan from a variant of sliced OT? To me, the purpose of sliced OT in general is improving the performance. The downside is that it doesn't yield an explicit plan or map unlike some other OT variants. Now, by using the proposed EST, we could both improve the runtime AND obtain an explicit plan. I assume that's the main contribution of this work. The paper shows the plan but I didn't find the improvement of runtime.

---

> > ### Author Response · Authors · 2024-11-25
> >
> > > Just a minor point. I would reverse the orders of the rows in Figure 4, so that rows with the best quality are closer to the original OT interpolation.
> >
> > This is a great suggestion, and we have updated the Figure in the paper accordingly.
> >
> > > I agree with "A metric offers a precise and consistent way to measure the "distance" between two elements in a space." My point was that the purpose of proving the metric properties in the paper was not clear. Why is it important that "the resulting transportation plan be used to define a metric between the two probability measures"? What if it's a pseudo metric? What applications of the plan will not hold if it doesn't define a metric? I understand the benefits of a true metric but I didn't find the benefits demonstrated in the paper.
> >
> > The reviewer raises a fantastic point.
> >
> > One of the main motivations for our paper originates from a statement highlighted in [Mahey, 2023](https://arxiv.org/pdf/2307.01770): although Sliced-Wasserstein-like distances are attractive due to their computational efficiency, proper metric properties, and ability to metricize weak* convergence, they do not provide an optimal transport (OT) plan as classical Optimal Transport theory does (i.e., when computing the Wasserstein distance, one also obtains the corresponding optimal coupling). Thus, one of our goals was to contribute along these lines by developing a technique that yields both a metric and a coupling between probability measures in $\mathbb{R}^n$. Furthermore, we proved that the new metric induces a topology equivalent to that of weak* convergence (in the case of discrete probability measures).
> >
> > Thus, we establish the metric property of our Expected Sliced Transport framework to ensure theoretical rigor and demonstrate that, while prioritizing computational efficiency in constructing the transport plan, we preserve the core theoretical benefits of optimal transport, which is its metricity.
> >
> > The OT framework simultaneously provides 1) an optimal transportation plan that establishes correspondences between the two input measures or between samples in empirical settings and 2) a metric that quantifies the distance between the input measures. Our proposed framework adheres to this paradigm, offering a transportation plan that defines a metric between the input measures. The reviewer raises an interesting question: beyond theoretical rigor, what are the practical advantages of obtaining a transportation plan that defines a metric rather than one that merely provides a pseudo-metric?
> >
> > The practical benefits are implicitly demonstrated in the classification experiments presented in the paper, as well as in the additional results provided during the rebuttal on ModelNet40. Consider the transportation plans obtained via entropy-regularized OT (using the Sinkhorn algorithm). In our classification experiments, we leveraged the core idea of Linearized OT to utilize the transportation plans for classification tasks. In this setting, where $\lambda$ is the entropy regularization coefficient, we observe the following behavior:
> >
> > - As $\lambda \to 0$, we recover the standard OT plans, which yield the highest classification accuracy.
> > - As $\lambda$ increases, the classification accuracy progressively declines.
> > - In the limit as $\lambda \to \infty$, the transportation plan approaches the outer product of the input measures, providing no useful information for distinguishing between classes.
> >
> > This demonstrates that the transportation plans maintaining metric properties (as achieved when $\lambda \to 0$) are crucial for preserving the discriminatory information needed for classification. Below, we reiterate the classification results reported in the paper, which support our argument.
> >
> > | Sinkhorn          | $\lambda_0=0$ (OT) | $\lambda_1>\lambda_0$ | $\lambda_2>\lambda_1$ | $\lambda\to\infty$ (Outer Product) |
> > | ----------------- | ------------------ | --------------------- | --------------------- | ---------------------------------- |
> > | Point Cloud MNIST | 96%                | 96%                   | 92%                   | 21%                                |
> > | ModelNet40        | 82%                | 79%                   | 66%                   | 6%                                 |
> >
> > Interestingly, a similar observation can be made for our proposed method as a function of number of slices. Our proposed method is a metric when we have expectation on slices, i.e., as $L\to\infty$. We see that our method performance also increases as we approach having a plan that defines a true metric.
> >
> > | ESP        | $L=1024$ | $L=128$ | $L=32$ | $L=4$  |
> > | ---------- | -------- | ------- | ------ | ---- |
> > | Point Cloud MNIST |    96%  | 94%     | 89%    | 84%  |
> > | ModelNet40 | 79%      | 77%     | 64%    | 12%  |
> >
> > These numerical results show the superiority of metric-inducing transport plans. We postpone a more rigorous analysis of applications that require metric-inducing transport plans for future work.

---

> > > ### Author Response · Authors · 2024-11-25
> > >
> > > > But we don't necessarily need explicit couplings to interpolate between the marginals, do we? We could solve a barycenter problem to get the interpolated measures. With the explicit couplings, yes, it's more convenient but I don't think it's required. So back to my original point, I was not arguing about the questions themselves. I was asking about the purpose of asking those questions. Given prior work on Sliced OT and OT in general, what is the purpose of an explicit plan from a variant of sliced OT? To me, the purpose of sliced OT in general is improving the performance. The downside is that it doesn't yield an explicit plan or map unlike some other OT variants. Now, by using the proposed EST, we could both improve the runtime AND obtain an explicit plan. I assume that's the main contribution of this work. The paper shows the plan but I didn't find the improvement of runtime.
> > >
> > > We agree with the reviewer that solving the barycenter problem (whether with fixed or free support) could produce similar interpolation results. However, as the reviewer noted, having access to the transport plan significantly simplifies the process. Specifically, for free-support interpolation, the optimization involved in solving the barycenter problem is more complex and less straightforward.
> > >
> > > We couldn't agree more with the reviewer's concise summary of our work: "the purpose of sliced OT in general is improving the computational performance. The downside is that it doesn't yield an explicit plan unlike some other OT variants. Now, by using the proposed EST, we could both improve the runtime AND obtain an explicit plan."
> > >
> > > Regarding the runtime improvement, in what follows we first provide a discussion on computational complexity of OT solvers and EST, and then provide a wall-clock time comparison with Sinkhorn.
> > >
> > > For the following analysis, let us consider two finite discrete probability measures on $\mathbb R^d$, $d>1$, concentrated at $n$ particles ($\mu=\sum_{i=1}^n p_i\delta_{x_i}$, $\nu=\sum_{j=1}^n q_j\delta_{y_j}$).
> > >
> > > The complexity of using the linear programming approach for solving the optimal transport problem between them is of order $\mathcal{O}(n^3\log(n))$.
> > >
> > > When using entropic regularization, the problem becomes easier to solve using iterative algorithms like Sinkhorn's algorithm, having a complexity of order $\mathcal{O}(n^2 \log(n)/\lambda^2)$, where $\lambda$ is the regularization parameter
> > > ([Dvurechensky et al., 2018](https://arxiv.org/pdf/1802.04367), [Altschuler et al., 2018](https://arxiv.org/pdf/1705.09634)), but introducing an approximation to the original problem.
> > >
> > > When dealing with measures in the real line, by using sorting algorithms, we reach a complexity of order $\mathcal{O}(n\log(n))$.
> > >
> > > The proposed EST method has following computational complexity when considering $L$ slices or unit vectors $\\{\theta_l\\}_{l=1}^L$ in $\mathbb{S}^{d-1}$:
> > >
> > > - For each one of the $L$ directions ($\\{ \theta_l \\}_{l=1}^L$), the slicing operation involves  $\mathcal{O}(Ldn)$ operations to obtain $\theta_l \cdot x_i $ and $\theta_l \cdot y_j $ for $i,j \in [1,n]$.
> > >
> > > - To compute the 1D plan $\Lambda_{\theta_l}^{\mu,\nu}$, for each $L\leq \ell\leq L$, we essentially need to solve a one-dimensional optimal transport problem (sorting problem), which is of order $\mathcal O(n\log(n))$. Thus, when considering all the slices, this step is of order $\mathcal O(Ln\log n)$.
> > > - The lifting process that gives rise to $\gamma_{\theta_l}^{\mu,\nu}$ does not requires operations to be taken into account.
> > > - The plan $\bar\gamma^{\mu,\nu}$ can be represented as an $n\times n$ matrix. The $(i,j)$-entry is given by $\sum_{l=1}^L\gamma_{\theta_l}^{\mu,\nu}(\{(x_i,y_j)\})$, requiring $L$ operations. Thus, the complexity of this step is $\mathcal O(L n^2)$.
> > > - Finally, once we have $\bar\gamma^{\mu,\nu}$,  for computing the EST-distance we require another $\mathcal O(n^2d)$ operations.
> > >
> > > Thus, the complexity of computing the plan $\bar\gamma^{\mu,\nu}$ is $\mathcal O(L(nd+n\log n+n^2))$, or simply, $\mathcal O(Ln^2+Lnd)$, and that of the EST-distance is $\mathcal{O}((L+d)n^2)$. As is generally the case, $n$ is much larger than $d$, so the computational complexity of both the EST plan and the EST distance is of order $\mathcal{O}((L+d)n^2)$.
> > >
> > >
> > > Besides, in the following Figures ([Fig.1](https://postimg.cc/kDvL4ySn) - [Fig.2](https://postimg.cc/N5GSrr8R)) we report the wall-clock times of the Sinkhorn algorithm, used for computing the entropic regularized version of OT (for different values of the regularization parameter $\lambda$), compared to the proposed EST distance (calculated with different numbers of slices $L$) between two $N$-sized empirical distributions. The only difference between the two figures is the scale used for the horizontal axis, which has been adjusted for visualization purposes.
> > >
> > > We have added this discussion and figures in the new Subsection 3.1 and in the Supplementary Material (Appendix C).

---

> > > ### Comment · Reviewer_Uqt2 · 2024-11-25
> > >
> > > Thank you.

---

### Official Review · Reviewer_A36p · 2024-11-08

**Soundness:** 3
**Presentation:** 3
**Contribution:** 3
**Rating:** 6
**Confidence:** 2

**Summary:**

They propose a novel method to give transport plans between disctrete distributions on Eucledian spaces named "expected sliced transport plan".
Although sliced Wasserstein distances in previous studies are computationally efficient in comparison to Wasserstein distance itself, they do not propose concrete transport plans between distributions.
To address this problem, they consider computationally efficient transport plans between distributions via slicing and lifting discussion.
It is shown that the proposed plans induce a metric for probability distributions.
In addition, they examine the performance of plans in several numerical experiments.

**Strengths:**

They succeed in proposing sliced transport plans inducing metric sfor probability measures.
As they claim, previous studies for sliced Wasserstein distances do not necessarily provide concrete transport plans.
While their results are only for discrete measures, I believe that it should be applicable to many problems in the real world.

In addition, they propose a tempered measure $\sigma$ on $\mathbb{S}^{d-1}$ in Section 3.2, which makes the performance of transport better.
This result gives motivation to consider general distributions on $\mathbb{S}^{d-1}$ and corresponding expected sliced transportation other than the uniform distribution.

**Weaknesses:**

While the motivation is computational efficiency, they do not argue it sufficiently. The discussion starting at Line 432 should be the only part referring to the computational complexities, but I believe that this is insufficient. Their claim is that the computational complexities of the proposed method does not change w.r.t. the temperature parameter $\tau$ while that of the entropic OT changes w.r.t. the regularization parameter $\lambda$. What the author(s) should compare here is not the complexities **within** the algorithms but those **between** the algorithms.

**Questions:**

On a very minor point: it can be better to unify the representations "transportation plan" and "transport plan".

---

> ### Author Response · Authors · 2024-11-23
> **Reply to Reviewer A36p's Feedback**
>
> Thank you for your time and thoughtful feedback. Below, we provide our response to your comments. We hope this discussion addresses your concerns and look forward to further engagement.
>
> ### Weaknesses
>
> > While the motivation is **computational efficiency**, they do not argue it sufficiently. The discussion starting at Line 432 should be the only part referring to the computational complexities, but I believe that this is insufficient. Their claim is that the computational complexities of the proposed method do not change w.r.t. the temperature parameter while that of the entropic OT changes w.r.t. the regularization parameter. What the author(s) should compare here is not the complexities within the algorithms but those between the algorithms.
>
> **Answer:**
>
> Line 432: As the temperature increases, the transport plan exhibits less mass splitting, similar to the effect of decreasing the regularization parameter $\lambda$ in entropic OT. However, unlike entropic OT, where smaller regularization parameters require more iterations for convergence, the computation time for expected sliced transport remains unaffected by changes in temperature.
>
> **Computational complexity:**
> For the following analysis, let us consider two finite discrete probability measures on $\mathbb R^d$, $d>1$, concentrated at $n$ particles ($\mu=\sum_{i=1}^n p_i\delta_{x_i}$, $\nu=\sum_{j=1}^n q_j\delta_{y_j}$).
>
> The complexity of using the linear programming approach for solving the optimal transport problem between them is of order $\mathcal{O}(n^3\log(n))$.
>
> When using entropic regularization, the problem becomes easier to solve using iterative algorithms like Sinkhorn's algorithm, having complexity of order $\mathcal{O}(n^2 \log(n)/\lambda^2)$, where $\lambda$ is the regularization parameter
> ([Dvurechensky et al., 2018](https://arxiv.org/pdf/1802.04367), [Altschuler et al., 2018](https://arxiv.org/pdf/1705.09634)), but introducing an approximation to the original problem.
>
> When dealing with measures in the real line, by using sorting algorithms, we reach a complexity of order $\mathcal{O}(n\log(n))$.
>
> The proposed EST method has following computational complexity when considering $L$ slices or unit vectors $\\{\theta_l\\}_{l=1}^L$ in $\mathbb{S}^{d-1}$:
>
> - For each one of the $L$ directions ($\\{ \theta_l \\}_{l=1}^L$), the slicing operation involves  $\mathcal{O}(Ldn)$ operations to obtain $\theta_l \cdot x_i $ and $\theta_l \cdot y_j $ for $i,j \in [1,n]$.
>
> - To compute the 1D plan $\Lambda_{\theta_l}^{\mu,\nu}$, for each $L\leq \ell\leq L$, we essentially need to solve a one-dimensional optimal transport problem (sorting problem), which is of order $\mathcal O(n\log(n))$. Thus, when considering all the slices, this step is of order $\mathcal O(Ln\log n)$.
> - The lifting process that gives rise to $\gamma_{\theta_l}^{\mu,\nu}$ does not requires operations to be taken into account.
> - The plan $\bar\gamma^{\mu,\nu}$ can be represented as an $n\times n$ matrix. The $(i,j)$-entry is given by $\sum_{l=1}^L\gamma_{\theta_l}^{\mu,\nu}(\{(x_i,y_j)\})$, requiring $L$ operations. Thus, the complexity of this step is $\mathcal O(L n^2)$.
> - Finally, once we have $\bar\gamma^{\mu,\nu}$,  for computing the EST-distance we require another $\mathcal O(n^2d)$ operations.
>
> Thus, the complexity of computing the plan $\bar\gamma^{\mu,\nu}$ is $\mathcal O(L(nd+n\log n+n^2))$, or simply, $\mathcal O(Ln^2+Lnd)$, and that of the EST-distance is $\mathcal{O}((L+d)n^2)$. As it is generally the case, $n$ is much larger than $d$, so the computational complexity of both the EST-plan and the EST-distance is of order $\mathcal{O}((L+d)n^2)$.
>
>
> Besides, in the following Figures ([Fig.1](https://postimg.cc/kDvL4ySn) - [Fig.2](https://postimg.cc/N5GSrr8R)) we report the wall-clock times of the Sinkhorn algorithm, used for computing the entropic regularized version of OT (for different values of the regularization parameter $\lambda$), compared to the proposed EST distance (calculated with different numbers of slices $L$) between two $N$-sized empirical distributions. The only difference between the two figures is the scale used for the horizontal axis, which has been adjusted for visualization purposes.
>
> We have added this discussion and figures in the new Subsection 3.1 and in the Supplementary Material (Appendix C).
>
> ### Questions:
> > On a very minor point: it can be better to unify the representations "transportation plan" and "transport plan".
>
> **Answer:** We thank the reviewer for addressing this. We have unified the terminology adopting "transport plan" everywhere.

---

> > ### Comment · Reviewer_A36p · 2024-11-25
> >
> > I appreciate the detailed response of the authors. The revised version represents the computational complexities of the algorithms clearly and addresses the concern properly. I have raised the score.

---

### Author Response · Authors · 2024-11-23
**Summary of Revisions**

We sincerely thank the reviewers and the area chair for their time, effort, and dedication to providing high-quality reviews.

We have addressed the reviewers' suggestions, provided further clarifications, and included new experimental results to enhance the paper. The main changes in the revised version are as follows:
 - We revised and restructured sections of the paper, including Figure 1 and its caption, to improve clarity.
 - We have added Subsection 3.1 to discuss the computational efficiency of the proposed EST approach, including a wall-clock time analysis comparing our method to Sinkhorn. Additional details are provided in Appendix C.
 - We added Appendix D to demonstrate further the efficiency of our proposed EST method for the classification task on a more complex dataset, ModelNet40 (Wu et al. CVPR 2015), using a Nearest Neighbor (NN) classifier.
 - We rewrote our first, second, and fourth contributions in the Introduction. In particular, in the first bullet, our intention is to give a summary of the technique we will elaborate on throughout the paper.

We have provided detailed responses to the reviewers' comments and look forward to engaging in a constructive dialogue.

---

### Meta-Review · Area_Chair_ciXA · 2024-12-21

**Metareview:**

The paper introduces the expected sliced transport, a new framework to construct transport plans between discrete probability measures using sliced optimal transport. The proposed method combines computational efficiency with explicit coupling between measures and defines a valid metric. Contributions include theoretical proofs, practical insights through experiments, and connections to the min-SWGG framework, advancing efficient transport-based machine learning techniques.

The paper effectively presented interesting ideas and adequately explained their usefulness. The reviewers pointed out potential improvements to the presentation and the need for additional analysis, which the authors adequately refuted.

**Additional Comments On Reviewer Discussion:**

A36p raised the point of insufficient discussion of computational costs; Uqt2 pointed out issues of extensibility of the proposed method and its applicability to real data; fUu9 pointed out the need for organization and intuitive presentation of the chapter and the need for analysis of computational costs. The authors appropriately refuted these points.

---

### Decision · Program_Chairs · 2025-01-22

Accept (Poster)